civil engineering/applied mathematics

community, wildland–urban interface, graph, vulnerability, risk

**Author for correspondence:**
Hussam Mahmoud
e-mail: hussam.mahmoud@colostate.edu

# Assessing wildland–urban interface fire risk

## Hussam Mahmoud and Akshat Chulahwat

Department of Civil and Environmental Engineering, Colorado State University, Colorado, CO 80523, USA

HM, 0000-0002-3106-6067

Recent wildfire events, in the United States (USA) and around the world, have resulted in thousands of homes destroyed and many lives lost, leaving communities and policy makers, once again, with the question as to how to manage wildfire risk. This is particularly important given the prevalent trend of increased fire frequency and intensity. Current approaches to managing wildfires focus on fire suppression and managing fuel build-up in wildlands. However, reliance on these strategies alone has clearly proven inadequate. As such, focus should be shifted towards minimizing potential losses to communities. Achieving this goal, however, requires detailed understanding of the factors that contribute to community vulnerability and the interplay between probability of ignition, vulnerability and calculated risk. In this study, we evaluate wildfire risk for four different communities across the USA for the duration of May to September to communicate a different perspective of risk assessment. We show, for the first time, that community risk is closely related to wind speed and direction, pattern of surrounding wildland vegetation, and buildings layout. The importance of the findings lies in the need for exploring unique viable solutions to reduce risk for every community independently as opposed to embracing a generalized approach as is currently the case.

## 1. Introduction

Wildfire events around the world in the last few years have resulted in astronomical social and economic losses. In the USA alone, the year 2018 has experienced the most catastrophic wildfire season on record in California, resulting in 7579 fires that burned a total of 1 667 855 acres (674 957 ha), the largest amount of burned acreage recorded in a fire season. The fires have caused more than $2.975 billion (2018 USD) in damages, including $1.366 billion in fire suppression costs. The year 2017, exhibited similar pattern of events both in frequency and intensity. An example is the Tubbs fire, which was named the most destructive fire before the Camp fire of 2018, and the Thomas fire, which caused over $2.2 billion in damages and $230 million in suppression costs. These events

are just a few examples of the level of destruction observed in wildfire events in the last couple of years. There are numerous other wildfire events that have left their mark on the world, including the Attica wildfires in Greece [1] and the 2019–2020 Australian fire, among others. The most surprising of all are the multiple fires in Jokkmokk, a small Swedish town in Lapland, which lies in the Arctic circle [2]. The recent devastating nature of these events is a testimonial to the fact that not only the intensity of wildfires are on the rise [3–5] but also the fire season is elongating as well [6,7]. The collective damages incurred in the USA due to wildfires in the year 2017 amounted to approximately US $18 billion. The United Nations (UN) Intergovernmental Panel on Climate Change (IPCC), comprising 192 nations, recently released the landmark report that forewarns the immediate need to take action to curb the rise in climate change. Within a few years, an overall 2°C rise in temperature is expected, which will result in a rise in wildfire events as well [8]. Therefore, there is a present need to devise mitigation strategies to reduce the impact of wildfires on communities.

Suppression and management of fuel build-up in wildlands has been one of the main tactics for lowering wildfire risk to communities, which alone has proven to be insufficient [9–12]. Wildfire mitigation is primarily focused on fire suppression and control [13]. However, other factors such as climate change, increase in community development near wildland–urban interface (WUI) areas [14], and rise in wildland density have resulted in a significant spike in high-intensity wildfires [10,15] and the associated expenditures [16,17]. Emphasis on managing public lands within and adjacent to communities provides some relief but falls short of the level of mitigation required to impact the susceptibility of communities to fire events [16]. Focusing on wildlands alone, without considering factors contributing to home ignition susceptibility, does not provide a complete picture of community vulnerability and risk to WUI fire.

At the community level, several wildfire protection programmes, such as Firewise, are aimed at informing residents of useful fire protection measures. These commonly include managing defensible spaces around houses, using fire retardant materials, and employing automatic fire suppression systems. While significant efforts are placed every year on increasing population awareness towards fire mitigation practices, no clear standardized policies exist. Current management practices focus primarily on control of wildfires in the wildlands, instead of focusing on the susceptibility of communities to the inevitability of wildfire exposure and establishing decisions based on calculated risks. In recognizing the major factors contributing to wildfire risk, a paradigm shift in wildfire management is required such that mitigation efforts are geared towards communities as well as the wildlands [10,18].

Most studies on wildfires are biased towards wildlands. There is paucity of literature on understanding wildfire propagation behaviour inside communities. Many researchers believe more attention should be paid to fire regulation needs, in addition to wildland management, for communities to coexist with nature. Calkin et al. [10] discussed this paradigm for controlling wildfire risk. Other researchers [13,16,19,20] have found that the characteristics of a home determine the ignition potential to a great extent as compared to its immediate surroundings. In addition, it has also been found that the housing arrangement within a community layout is a critical factor governing the likelihood of house ignition [21]. There are other factors as well, pertaining to built environment properties, such as housing density, fuel load and moisture, weather and some others [22]. As such adequate metrics are required to quantify the ignition potential, vulnerability and risk of fire damage to individual homes within a community so that informed mitigation decisions can be made, both by the authorities and home owners.

Determining vulnerability and risk of wildfire damage to homes and providing better understanding of the factors governing wildfire behaviour requires the use of suitable analytical and numerical tools. Computational fluid dynamics (CFD) models have been found to be the most effective for modelling wildfire propagation, since they are based on physics of the problems as opposed to semi-physics or empirical methods. However, CFD models are computationally very expensive and their use in very large problems (i.e. community-level analysis) is currently not feasible. With advances in computational infrastructure, in the near future, the use of CFD models will become a reality. However, with the risk of WUI fires on an astronomic rise each year, communities cannot afford to wait for the computational technology to match the complexity of the problem. With this in mind, the pressing need lies in exploring alternative directions for quantifying and studying wildfire risk to communities. In a previous paper [23], we proposed and showed that application of traditional graph theory concepts can provide a good understanding of the complexities involved in WUI fires.

Every natural hazard has certain characteristics, based on which their risk is quantified and communicated. For earthquakes, risk is communicated through magnitude of the shaking, hurricanes by wind velocity and storm surge, floods by measured depth of water and so on. For wildfires, researchers have developed detailed frameworks to quantify the potential of fire spread in wildlands; however, there is currently no standardized method of risk assessment or communication that can be

applied nationwide to WUI communities [24]. Wildfires are a natural phenomenon, similar to other natural hazards. Only recently (spring 2018), the Wildfire Disaster Funding Act (H.R. 2862) passed by Congress has classified wildfires as a natural hazard just like hurricanes, floods and earthquakes [25]. An abundance of knowledge exists on risk quantification of other hazards. In the case of earthquakes, we understand the underlying factors that govern infrastructure damage potential quite well. For instance, the depth of focus from the epicentre, the fault rupture mechanism, and the proximity of the infrastructure to fault lines are all factors that affect the damage potential. Damage potential from wildfires lack the same level of details as other hazards.

Even though wildfires can be classified as a natural disaster, they do not necessarily behave in a similar fashion as other hazards. Wildfire is the only natural hazard in which the intensity of the hazard increases with time in proportion to the volume of damage caused. The ignitable structures act as fuel to the ongoing fire and result in an increase in spread. In addition, embers from burning vegetation and combustible materials contribute substantially to fire initiation and spread in communities. Wildfire propagation is indeed complex and devising effective policies to reduce fire damage in communities requires shift in research attention towards addressing an unanswered question—have we quantified the underlying factors involved in wildfire events and do we understand their importance relative to each other? In this study, we attempt to answer this question using a graph model by carefully studying the factors contributing to vulnerability and risk to WUI fires in four distinct communities in the USA. We conduct the analysis for the months of May to September, representing a typical fire season, based on information collected from existing wildfire and weather databases. Using the analysis results, we draw out meaningful correlation patterns between wildfire risk and other underlying factors. We clearly show that risk depends on the community being evaluated and therefore should be communicated as such.

# 2. Material and methods

## 2.1. Risk framework

Several researchers have looked into quantifying risk of communities to wildfires [26–29]; however, comprehensive theoretical frameworks are lacking. Available frameworks only account for risk from the perspective of wildlands and do not take into consideration the susceptibility of communities based on their individual characteristics. A comprehensive definition of wildfire risk entails assessment of two key components—(i) probability of a wildfire event, and (ii) susceptibility of highly valued resources and assets to wildfire [10]. Under the framework devised in this study, community risk is reclassified into three stages of wildfire—(i) probability of wildland ignition ($P(Z(t))$), (ii) probability of wildfire that started in wildland to reach a specific WUI ($P(Y(t)|Z(t))$), and (iii) susceptibility of community provided that a wildfire reached the WUI ($P(X(t)|Y(t))$). Using these three stages, the net risk of a community ($R(t)$) for a particular day $t$ can be assessed using equation (2.1).

$$R(t) = P(Z(t) \cap Y(t) \cap X(t)) = P(Z(t)).P(Y(t)|Z(t)).P(X(t)|Y(t)). \tag{2.1}$$

Susceptibility of a community can be defined as the mean probability of fire reaching a house and causing ignition from the boundary of surrounding WUI. Therefore, the risk can be defined as the mean probability of fire reaching a house from the initial ignition point in the wildlands. The general definition of risk for any hazard is characterized by three components—(i) hazard, which is defined as the temporal probability of occurrence for a hazard of a particular intensity, (ii) vulnerability, which is defined as the degree of exposure, and (iii) amount of elements at risk, which is the quantification of exposed elements. The terms vulnerability and elements at risk are coupled to form the vulnerability term. This is because for other hazards, a typical prototype structure has a distinct value for vulnerability (probability of failure, obtained from fragility functions). For example, a moment frame on soil type D subjected to specific earthquake excitation will perform the same way regardless of its location (i.e. as long as the building is the same and the load is the same, it does not matter where the building is placed). For wildfire, the vulnerability of a specific type of building will vary depending on its location and orientation within the community.

Since the focus of this study is towards community-specific risk, the probability of fire reaching the interface, once initiated, is assumed to be one for all cases ($P(Y(t)|Z(t)) = 1$). This also provides a more conservative value of risk. The probability of ignition for each community is derived based on the National Fire Danger Rating System (NFDRS), as discussed in subsequent sections. The susceptibility of a community is defined as the mean vulnerability of all ignitable components within

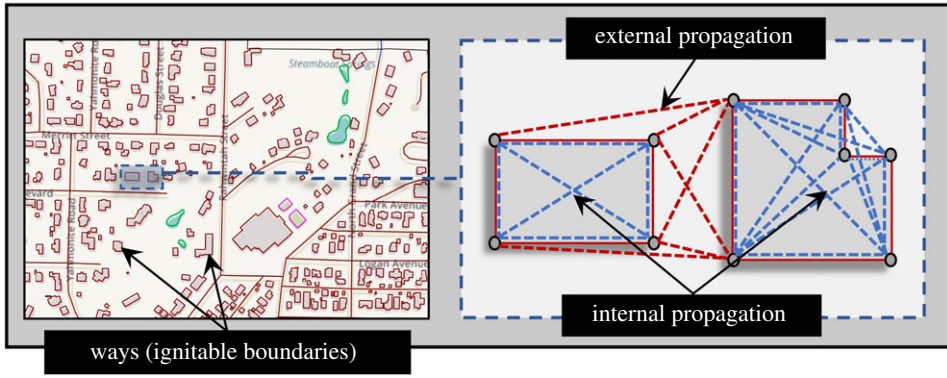

**Figure 1.** Sample graph representation of a community showing different types of propagation (Map data Copyright © OpenStreetMap contributors [35]).

the community. The mean is an acceptable performance metric for the scope of this study; however, it can be replaced by a weighted mean such that the weights for each ignitable component corresponds to its importance in a community.

## 2.2. Community wildfire propagation model

When a wildland fire enters a community it is referred to as a WUI fire. The underlying propagation mechanisms for these fires are identical to wildland fires; however, the difference in topographic features creates explicit differences in behaviour. Computationally, efficient models for fire propagation have been explored in the past using concepts of graph theory, both for wildlands [30–32] and urban settings [33,34]. The urban fire problem can be formulated analogous to a network flow problem in graph theory. In a previous study [23], a quasi physics-based graph model (AGNI-NAR: *Asynchronous Graph Nexus Infrastructure for Network Assessment of Wildfire Risk*) was used to evaluate vulnerability of Oakland, California to wildfire. The use of the model is extended in this study to assess risk of different communities to wildfires. A graph network is developed based on the geographical data of each community. The ignitable areas of a community are first identified then classified based on their intrinsic susceptibility to ignition. The ignitable areas are identified based on the list of classification shown in electronic supplementary material, table S1. A suitable directed graph is developed using propagation probabilities of different modes of heat transfer between ignitable ways of the community. There are four different modes of heat transfer considered in this study—(i) conduction, (ii) convection, (iii) radiation, and (iv) embers. Each ignitable component of a community is defined by a set of nodes that define its boundary. This boundary is referred to as a 'way', as shown in figure 1. The figure shows how each way (house) is segregated into a set of nodes to represent its outer boundary, followed by edges between the nodes to represent the interaction between them. The interaction between nodes is classified into two types, based on the nature of source and target nodes, as (i) internal, and (ii) external propagation, such that the internal propagation is defined by heat transfer within nodes of the same way and external as the heat transfer between nodes of different ways.

Both internal and external propagation are governed by different modes of heat transfer [20,36]. When nodes $i$ and $j$ belong to the same way, the ignition transfer probability is given by conduction probability only, $P_{\text{cond}}^{(i,j)} \in \{0, 1\}$. The focus of this study is primarily on the community level, hence modelling of internal propagation is simplified. A test is conducted to show that the effect of modelling internal propagation on the community as a whole would be minimal. The results of the test for the four communities are shown in electronic supplementary material, figure S1. Internal probabilities are assumed to be one $P_{int}^{(i,j)} = 1$ for all cases to obtain the most conservative estimates from the analysis. External propagation comprises primarily three components—(i) convection, $P_{\text{conv}}^{(i,j)} \in \{0, 1\}$; (ii) thermal radiation, $P_{\text{rad}}^{(i,j)} \in [0, 1]$; and (iii) ember spotting, $P_{\text{ember}}^{(i,j)} \in [0, 1]$, which accounts for majority of fire damage in WUI fires [19]. The total probability of external propagation is defined by equation (2.2) and the effective probability of transfer between nodes is defined by equation (2.3). More details on the different propagation mechanisms can be found in [23].

$$P_{\text{total}}^{(i,j)} = (P_{\text{conv}}^{(i,j)} \cup P_{\text{rad}}^{(i,j)} \cup P_{\text{ember}}^{(i,j)}) \tag{2.2}$$

and

$$P_{tr}^{(i,j)} = \begin{cases} \min(P_{total}^{(i,j)}, 1) & \text{if } \{j \in \mathcal{W}_{(m)}: i \notin \mathcal{W}_{(m)}\}_{m \in \mathbb{Z}} \\ P_{int}^{(i,j)} & \text{if } \{j \in \mathcal{W}_{(m)}: i \in \mathcal{W}_{(m)}\}_{m \in \mathbb{Z}}. \end{cases} \quad (2.3)$$

Once the weights of each edge ($P_{tr}^{(i,j)}$) are defined for the formulated directed graph, the vulnerability of each way is calculated as the mean probability of most probable paths (MPP) from a particular ignition source ($s$). The ignition source is defined as the first node in the graph to be activated by fire in the wildland. The position of this node can be on the WUI or even inside the community. The mean probability $P_m^{(s)}$ of propagation along a MPP is defined as the product of the edge weights (equation (2.4)).

$$P_m^{(s)} = \frac{1}{K} \sum_{x=1}^{K} \left[ \prod_{(i \to j) \in \mathcal{M}_{(x)}} P_{tr}^{(i,j)} \right], \quad (2.4)$$

where, $\mathcal{M}_{(x)}$ is the adjacency list of $x$ MPP given by $\mathcal{M}_{(x)} = \{(n_{(1)} \to n_{(2)}), \ldots, (n_{N_{(\mathcal{M}_{(x)})}-1} \to n_{N_{(\mathcal{M}_{(x)})}})\}$, $N_{\mathcal{M}_{(x)}}$ is the total members in adjacency list $\mathcal{M}_{(x)}$. The mean probability $P_m^{(s)}$ is averaged over $K$ MPPs (equation (2.4)). In this study, $K = 10$ for all analysis, since such value was sufficient to achieve convergence.

Parts of four different communities—(i) Austin (Texas), (ii) Jackson (Wyoming), (iii) Oakland (California), and (iv) Steamboat Springs (Colorado), from the USA, are chosen for conducting the risk analysis. The layouts of the four communities are shown in figure 2. The communities are selected due to their close proximity to wildlands and differences in their layouts. The number of nodes and ways identified in each community to formulate their corresponding graph networks are shown in electronic supplementary material, table S2. Each community has a unique footprint attributing to structure density, community layout, and vegetation distribution. Since the focus of this study is to draw out a comparison between the selected communities, certain assumptions are considered. All houses (referred to as 'ways') in each community are assumed to be identical in nature i.e. possess same material properties. Furthermore, the vegetative fuel present in each community is assumed to be of the same type.

## 2.3. Wildland ignition probability

For this study, the Wildland Fire Assessment System (WFAS), also known as the National Fire Danger Rating System (NFDRS), developed by the United States Forest Service (USFS) [37] is used. The WFAS performs daily fire danger forecasts with data from the National Digital Forecast Database for different locations across the USA. The fire danger forecasts result in rating levels that take into account current and antecedent weather, fuel types, and both live and dead fuel moisture. It primarily uses two performance indices—(i) burning index (BI) and (ii) energy release component (ERC). Assigning the fire danger index reflects staffing levels and climatological class breakpoints. Staff class represents the max/min fire danger rating of a location by assigning percentile values for the performance index selected (i.e. BI or ERC) for a specific day. This fire danger rating is then used to calculate an ignition probability. Specifically, linear interpolation is used to determine probability of ignition ($P(Z)$), as given by equations (2.5), (2.6) and (2.7).

$$P(Z) = \frac{1}{100}(m(d).I(d) + c(d)) \quad (2.5)$$

$$m(d) = \frac{(v_u^k(d) - v_l^k(d))}{(p_u^k(d) - p_l^k(d))} \quad (2.6)$$

and
$$c(d) = v_u^k(d) - m(d).p_l^k(d), \quad (2.7)$$

where $p_u^k(d)$ and $p_l^k(d)$ are appropriate upper and lower percentiles allotted on a daily basis by station managers and $v_u^k(d)$ and $v_l^k(d)$ are the performance indices values corresponding to the percentiles selected.

## 2.4. Effect of wildlands

Wildfires enter communities from the wildlands primarily through—(i) the WUI and (ii) embers generated from trees and vegetation in the wildland travelling by wind and landing inside the community. These two different mechanisms result in multiple ignition source nodes at the boundary and inside the community [36,38,39]. By considering fire paths from all possible source nodes, the

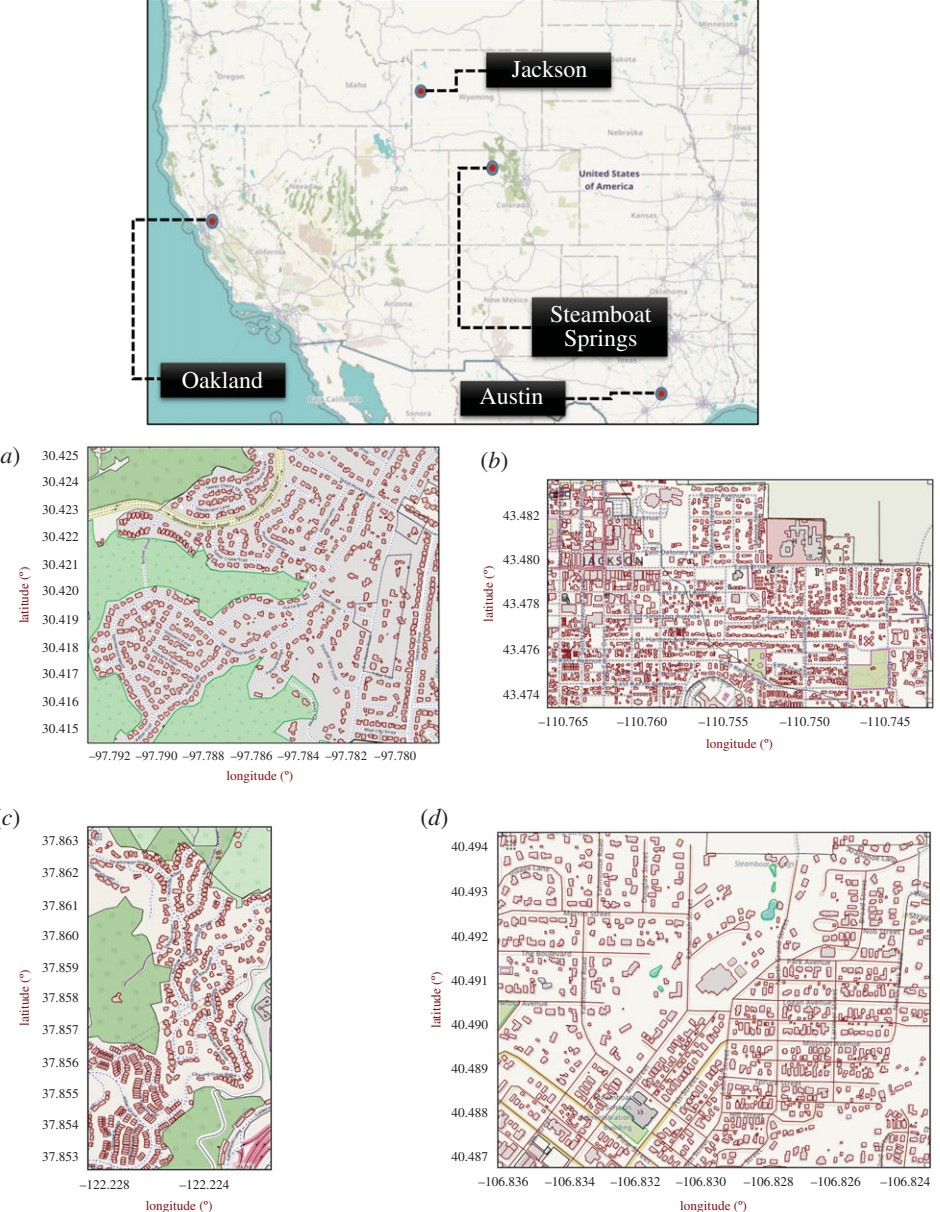

**Figure 2.** Community layout maps for (*a*) Austin (Texas), (*b*) Jackson (Wyoming), (*c*) Oakland (California) and (*d*) Steamboat (Colorado) (Map data Copyright © OpenStreetMap contributors [35]).

effect of wildlands on community vulnerability is accounted for. The total vulnerability ($V^{(z)}$) of destination node $z \in \mathcal{W}_{(m)}$ is calculated by equation (2.8).

$$V^{(z)} = \max \left( P_i^{(s)} \cdot P_m^{(s)} \right)_{\{s \in \mathcal{S}\}}, \tag{2.8}$$

where $\mathcal{W}_{(m)}$ is the way $m$, $P_i^{(s)}$ is the ignition probability of source node $s$, and $N_{\mathcal{S}}$ is the total number of source nodes in node set $\mathcal{S}$. The probability of ignition for each source is correlated to wind conditions and wildland vegetation in the vicinity of the community. The probability of ignition $P_i^{(s)}$ is defined by equation (2.9) and is function of wind direction ($\theta$), edge angle ($\phi^{(b,s)}$) and distance ($d^{(b,s)}$) between nodes $b$ and $s$, where node $b$ is one of the boundary nodes of the adjacent wildland. The function is based on the ember model developed by Martin and Hillen [40], which follows the concept of birth-jump processes and has been studied in the context of wildfire spotting [41].

$$P_i^{(s)} = f(d^{(b,s)}, \phi^{(b,s)}, \theta). \tag{2.9}$$

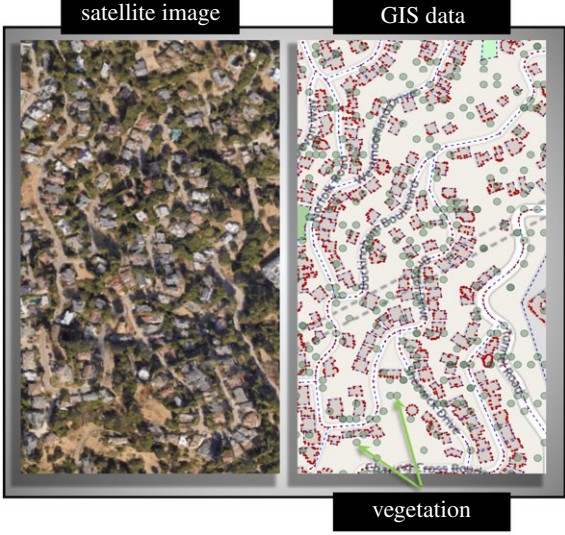

**Figure 3.** A part of Oakland community with additional nodes depicting vegetation within the community.

## 2.5. Modelling vegetative fuel

Each community has certain amount of vegetative fuel present within its vicinity. Vegetation around houses tend to increase the exposure to ignition, as also shown in recent studies [42,43]. To account for this effect in the vulnerability calculations for each house in a community, the vegetation is modelled exclusively into the fire propagation framework. The GIS data for community layouts is derived from Openstreetmap in this study; however, it does not provide details on locations of discrete vegetation within communities. Satellite images from Google Earth are therefore used to fill the information gap on vegetation. To incorporate the effects of vegetation into the wildfire model, individual vegetation are introduced into the framework in the form of separate nodes. These nodes are added to the graph networks formulated for each community. However, by adding these additional nodes the computational requirements for the analysis increases substantially. From the satellite image of each community it is clear that the vegetation in each case is almost uniformly distributed close to the houses. In light of this observation, it can be hypothesized that effect of vegetation can be modelled without the additional nodes by combining the volume of vegetative fuel with that of houses (or ways) in close vicinity. An array of analysis is conducted on all four communities to test the hypothesis.

Two approaches are tested in this study for modelling vegetation within a community—(i) explicit vegetation modelling, and (ii) simplified vegetation modelling. The first method pertains to modelling vegetative fuel as a set of explicit nodes, while the latter pertains to modelling vegetation by combining their fuel volume with other ways. Satellite images for the four selected communities are taken from Google Earth to obtain spatial information of vegetative fuel within the selected communities. An example image for the community of Oakland is shown in figure 3. Vegetation inside the community is modelled as a collection of individual nodes with each node representing an individual vegetation entity. Using the satellite images of each community, the mean vegetation per unit square area is used to formulate uniform distributions of vegetation for the communities. The distributions are used to generate vegetation nodes on layout of each community. The ignitable ways within community of Oakland along with randomly generated vegetation is also shown in figure 3.

Multiple random spatial configurations of vegetation layout are generated for each community layout. The vegetation nodes are then assigned individual properties separately. For each configuration, properties of individual vegetation nodes, such as diameter and height, are decided by the distributions given in electronic supplementary material, table S3. To assess the difference in accuracy between the two methods, both are tested on the four selected communities. The wind speed and direction for the tests are chosen based on wind conditions of 1 May 2007 for each location. For the explicit vegetation modelling method, a total of 100 different vegetation configurations are generated for each community layout. The mean vulnerability is evaluated for each configuration using the wildfire propagation model. To account for the additional vegetation nodes, no changes are made to the propagation model framework. For the simplified modelling method, the mean vegetative fuel volume per unit area is

calculated from the satellite images and added to each of the houses (or ways). Vulnerability analysis for the two methods shows strikingly similar results. The difference in vulnerability values for the two methods are observed to be within 10% bounds. The results are shown in electronic supplementary material, table S4. For all analysis in this study, the vegetation are not separately included in the vulnerability calculations to keep the processing time manageable. This does not necessarily mean that the proposed hypothesis would hold true always. In the case of communities with non-uniform layout and fuel density, the vegetation might have to be modelled separately.

## 2.6. Modelling fire intervention

The effect of active and passive fire mitigation is incorporated into the graph model using a static intervention framework. The intervention framework is incorporated to model the resistance provided by the communities, which entails fire mitigation efforts by firefighters and private home owners. An intervention strength $\mu$ is first selected, which represents a percentage of ways (or houses) affected by mitigation measures in a community. Based on the strength factor, a percentage of the total ignitable ways present in the community $\mu.N_W$ are chosen at random, assuming a uniform distribution. A new set is created $\{\mathcal{W}_M | \mathcal{S} \notin \mathcal{W}_M\}$, where $\mathcal{S}$ is the set of source nodes. The set $\mathcal{W}_M$ is formulated such that it does not contain any source nodes. The inflow and outflow for each node of the ways in the formulated set $\mathcal{W}_M$ are altered by changing the indegree and outdegree to modify the original graph ($G$), as given by equation (2.10), where $a_{(v,j)}$ is the weight of the edge between nodes $v$ and $j$.

$$a_{(v,j)} = \alpha.a_{(v,j)} \quad a_{(j,v)} = \beta.a_{(j,v)} \; \forall \; \{v \in \mathcal{W}_M^{(l)} | l = 1:N_{\mathcal{W}_M}, j = 1:n\}, \tag{2.10}$$

where $\alpha$ and $\beta$ are mitigation scaling factors that are assumed to be 0.10 and 0.75. $\alpha$ represents the scaling factor for outflow from node $v$, which would be affected by factors such as sprinkler systems, among others. $\beta$ represents the scaling factor for inflow to node $v$, which would be affected by individual house properties such as roofing, siding material, among others. The value of $\alpha$ is assumed based on the fire mitigation capacity of sprinklers [44]. Depending on the location of ways chosen for intervention, the vulnerability of community changes [23]. Some details on the parameters controlling the scaling factor and its role in formulating the fire intervention framework is discussed in electronic supplementary material, S4. For this study, only one configuration (chosen at random) is used for each community and the intervention strength is kept a constant at 50%, i.e. for only half the houses in each community intervention is applied.

## 3. Results

For all four communities, an intervention strength of $\mu = 50\%$ is used for all cases, which indicates 50% of all ways in the communities are altered to introduce the effect of fire mitigation. The risk of each test community is calculated each day for the months of May to September. Figures 4–7 show the daily risk values for each of the five months for the years 2007, 2012 and 2017, for the four selected communities. Mean risk values ($R_m$) for each month are shown in the figures, which are defined as the average of risk for each day ($R(t)$), as given by equation (3.1).

$$R_m = \frac{1}{t} \int_0^t R(t) \, dt. \tag{3.1}$$

The risk values for each day of the months are calculated based on the daily wind and ignition probability data (electronic supplementary material, S5). The calculated daily ignition probability values are shown in electronic supplementary material, figures S8–S11 and the daily vulnerability values are shown in electronic supplementary material, figures S12–S15 for each community. Based on the risk patterns observed for all communities (figures 4–7), it is evident that even if the chance of a wildfire ignition is high enough it may not necessarily result in high risk for communities. In several cases, however, high vulnerability, caused by unfavourable wind, is shown, leading to higher risk. Some of the most destructive wildfires in history were accompanied by strong seasonal winds—(i) the Oakland wildfire (1991) by El Diablo winds [45], (ii) the Thomas fire (2017) by Santa Ana winds [46], and (iii) Australia bushfires by Foehn winds [47]. The graph model is formulated in a way so as to allow for incorporating all types of wind conditions ranging from mild to extreme events. For a given spatial resolution, the effective nodal probabilities ($P_{tr}(i, j)$) can be updated based on the wind field pattern observed.

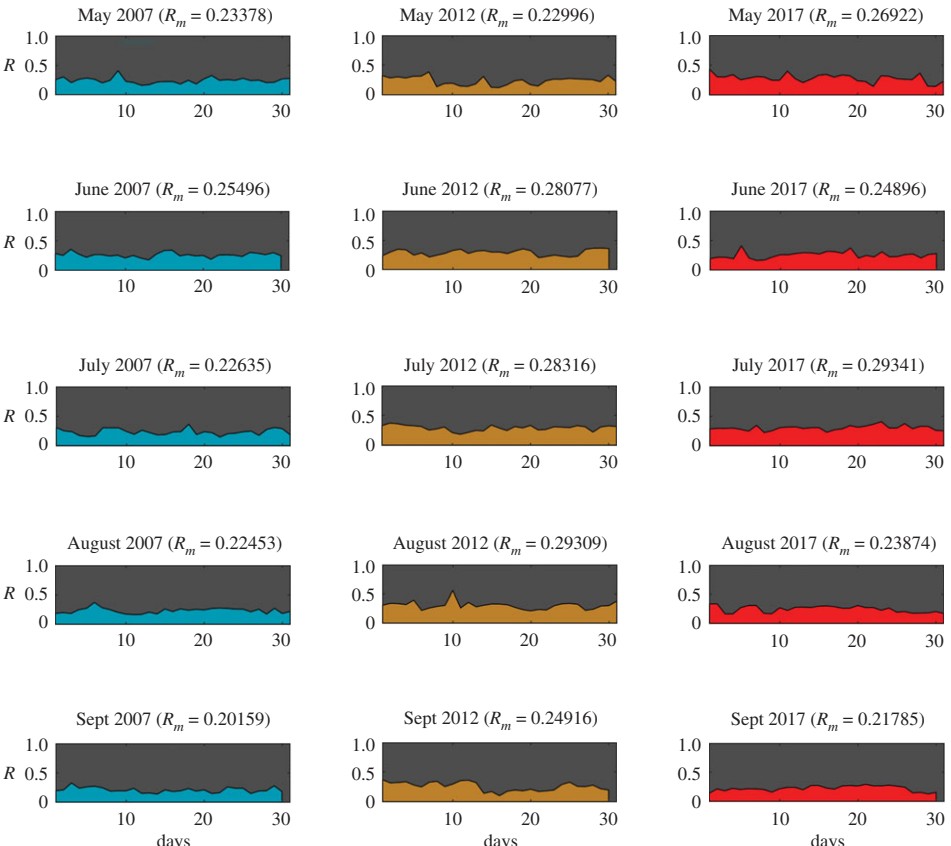

**Figure 4.** WUI fire risk in months May–September for years 2007, 2012 and 2017 for Austin (Texas) (Map data Copyright © OpenStreetMap contributors [35]).

For varying wind events no changes are required in the model formulation; however, the temporal resolution of the analysis would have to be increased. For instance, hourly wind data could be used instead of daily wind data. Based on the mean risk observed for different months of each community, Jackson is observed to have the highest overall risk (figure 5) and Steamboat to have the lowest (figure 7), while Austin (figure 4) and Oakland (figure 6) showed intermediate risk relative to the other communities.

Steamboat exhibited both lower ignition probabilities (electronic supplementary material, figure S11) as well as lower community vulnerability (electronic supplementary material, figure S15). The latter is specifically lower for Steamboat due to the absence of significant wildland vegetation in the vicinity of the community, which limited the entry points for the wildfire. The discontinuous layout of the community further reduced the vulnerability. As noted before, wildfire risk is function of both wildland ignition probability and community vulnerability. To understand the correlation of risk with these two key parameters, the Pearson correlation coefficient is calculated for each community separately (table 1). For all communities, except Jackson, the correlation is observed to be stronger between risk and community vulnerability than risk and wildland ignition. This suggests that wildfire risk can be better regulated for these communities by controlling the community vulnerability. However, in the case of Jackson the correlation between risk and wildland ignition is higher, which suggests that risk in this case is primarily governed by the wildlands. A different perspective would be that it would require much more effort to bring down the risk for Jackson below a certain threshold, since it is situated in a high fire vulnerability region. Hence, the risk can be quantified based on two types—communities where wildfire risk can be regulated with more measures in the community and those where wildlands should be the focus.

As discussed earlier, wind conditions have a severe effect on the vulnerability of communities. Wind speed has a direct correlation to wildfire intensity, as observed from wildfire cases over the years. The effect of wind direction, on the other hand, is not so straightforward as it depends on the community layout. There might be multiple favourable directions of wind that might accentuate wildfire intensity. Polar fragilities for the test communities are calculated to show their respective sensitivity to wind direction (figure 8). These fragilities are formulated by varying the wind direction at an interval of

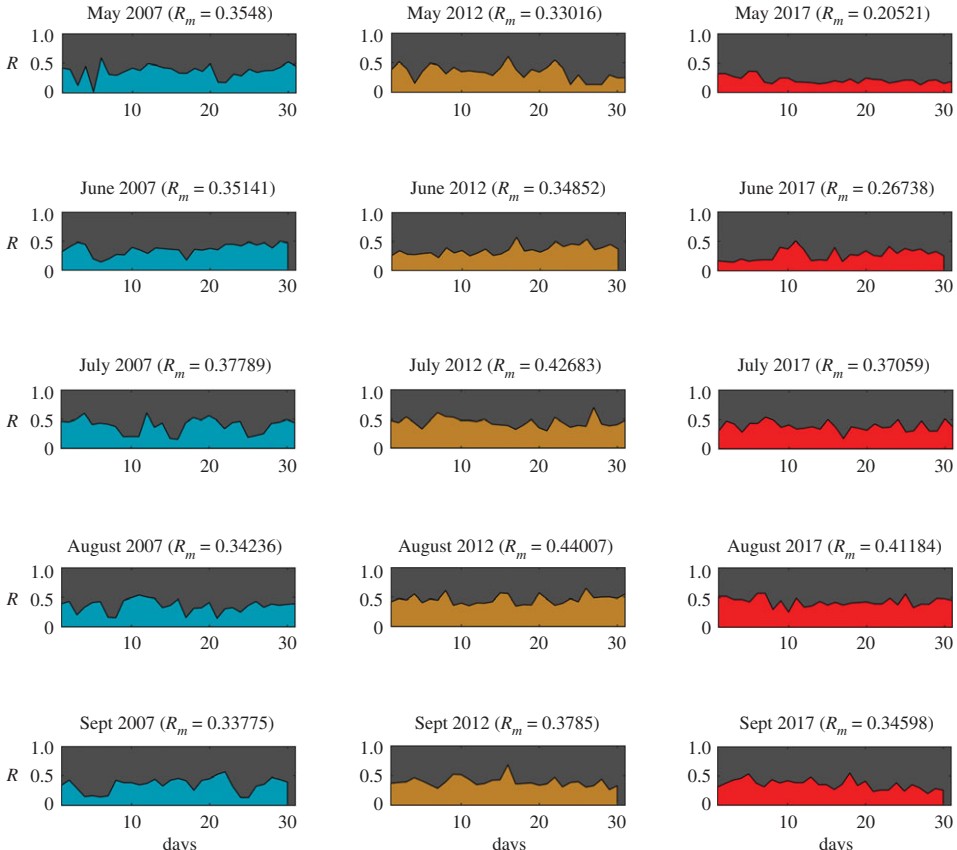

**Figure 5.** WUI fire risk in months May–September for years 2007, 2012 and 2017 for Jackson (Wyoming) (Map data Copyright © OpenStreetMap contributors [35]).

30° and calculating the mean vulnerability of the community. Wind direction is measured anticlockwise from the positive x-axis, such that a N–S wind is represented by $\theta = 270°$ and S–N wind by $\theta = 90°$. The individual data points in the polar fragilities represent the mean probability of fire reaching a house in the community from the WUI for a particular wind direction. For each wind direction, a new graph is formulated by updating the nodal probabilities, followed by the MPP calculation to determine mean community vulnerability. Similar fragility curves for different communities suggest uniformity across community layout, both in terms of fuel density and material property. Dissimilar fragility curves would suggest bias in certain directions. For each community, the dominant wind directions are observed to be different, which is to be expected given the different layouts of the four communities tested (figure 2). Jackson is observed to have the least effect of wind direction, while Oakland is observed to have the most.

During the 1991 Oakland wildfire, the situation was worsened by the seasonal Diablo winds which entered Oakland from the dominant direction shown in fragility curve ($\theta = 240°$). The mean vulnerability for the 1991 wildfire is calculated to be 0.79. In September 2017, a fire ignited in the same location in wildlands of Oakland as it did in the infamous 1991 Tunnel fire [48]. However, this time the fire agencies were able to suppress the fire before it reached the internal parts of the community due to the controlled wind conditions and prompt actions of the firefighters. The risk is observed to be 0.25, which represents 68.3% reduction in vulnerability just due to the absence/presence of certain wind conditions. Ideally, the difference in building material used for reconstructing Oakland after the 1991 wildfire would also have an impact on reducing the vulnerability of the community. However, due to data unavailability, it is assumed that the materials for all homes are the same as those used prior to the 1991 wildfire. Hence, the analysis presented does not account for reduction in risk due to changes in materials.

# 4. Discussion and conclusion

It has been more than a decade since the devastation caused by the 1991 Oakland fire; yet even today, we lack the means to manage such an event. In 2017 and 2018 alone, similar wildfire incidents have plagued

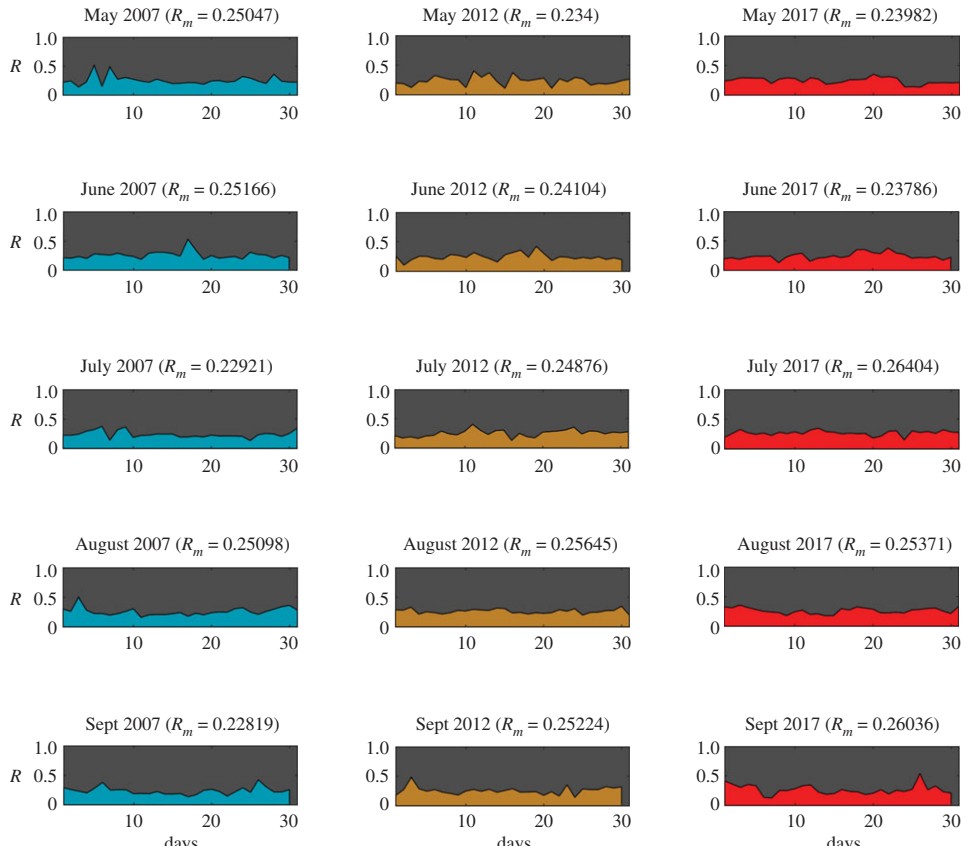

**Figure 6.** WUI fire risk in months May–September for years 2007, 2012 and 2017 for Oakland (California) (Map data Copyright © OpenStreetMap contributors [35]).

**Table 1.** Pearson correlation values between risk ($R$), vulnerability of community ($V$) and probability of wildland ignition ($P_i$), for each community

| correlation | Austin | Jackson | Steamboat | Oakland |
|---|---|---|---|---|
| $R$ and $P_i$ | 0.5886 | 0.8304 | 0.5158 | 0.3075 |
| $R$ and $V$ | 0.8407 | 0.3601 | 0.9534 | 0.8572 |
| $V$ and $P_i$ | 0.0878 | − 0.1781 | 0.2409 | −0.14 |

different parts of the world. Given the rise in temperature due to climate change, among other factors, this trend of wildfires is expected to increase, both in frequency and severity. In light of the rising potential risk to communities, effective strategies for wildfire management are required. Current strategies mainly entail fire suppression and fuel management in wildlands. Mitigation strategies geared towards complete containment of wildfires within the wildlands are nothing short of unrealistic. Limited information exist regarding the interplay of communities and wildfires. Unlike other hazards, for which there exists significant knowledge base, quantification of WUI fires is still a question for us. To better understand what factors govern the impact of WUI fires, tools to assess and quantify the risk to communities are required. We recognize that a move towards devising community-level mitigation strategies has been a recent focus, as evident by the International Wildland-Urban Interface Code (IWUIC), which pertains to minimum standards and requirements for location of buildings as well as defensible space, and materials and methods of construction. Further refinements of such code, or any other similar code, require detailed quantitative assessment of risk of vulnerable communities.

In this study, we evaluated wildfire risk for four different communities around the USA. Using local wind data, community buildings layout and the probability of ignition in the wildland, we calculated risk for every community. We showed that risk is community-specific and is a function of different

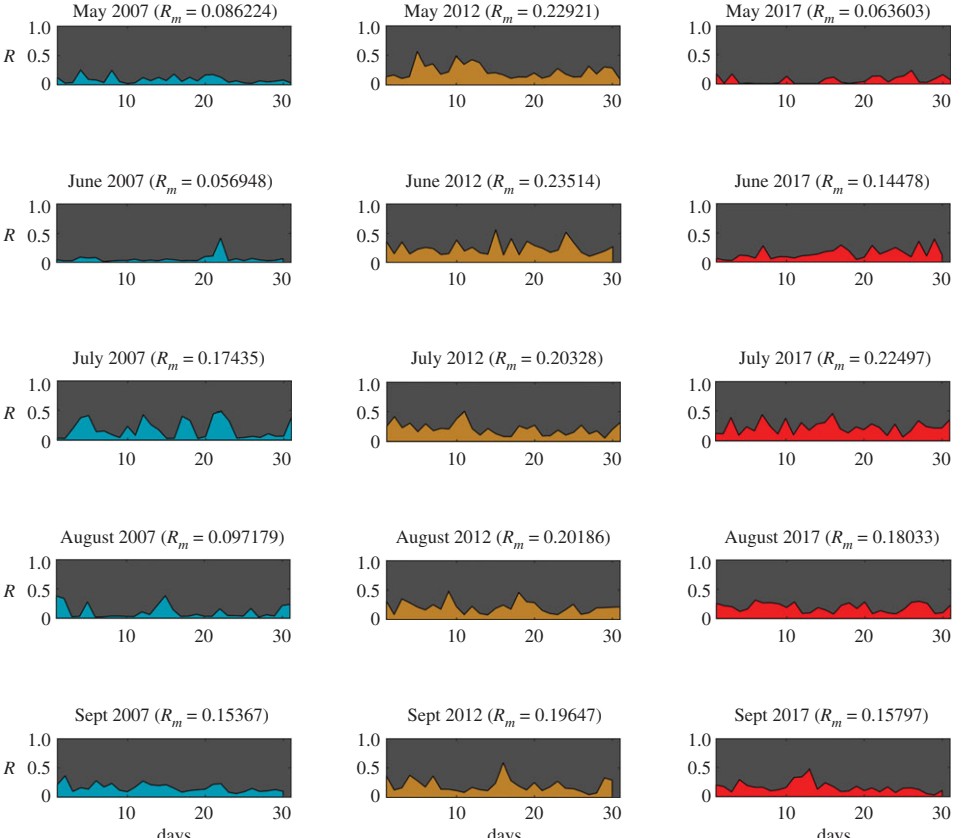

**Figure 7.** WUI fire risk in months May–September for years 2007, 2012 and 2017 for Steamboat Springs (Colorado) (Map data Copyright © OpenStreetMap contributors [35]).

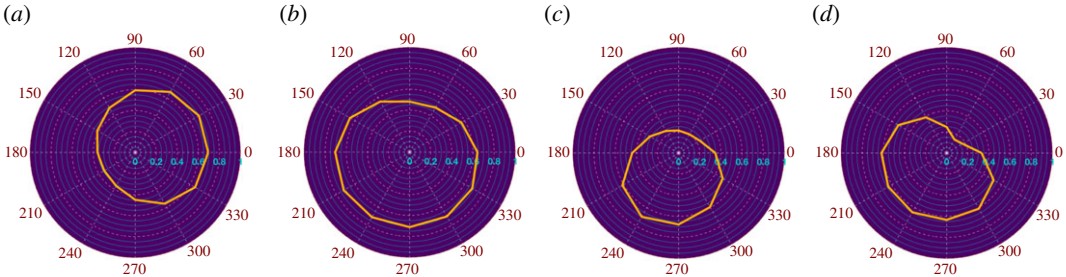

**Figure 8.** Polar fragilities for (*a*) Austin (Texas) (*b*) Jackson (Wyoming) (*c*) Oakland (California) (*d*) Steamboat (Colorado) at wind speed $V_w = 15 \text{ m s}^{-1}$ (Map data Copyright © OpenStreetMap contributors [35]).

environmental, location, and layout parameters. A recently published study [49] found that California, Texas and Colorado experienced highest building losses due to wildfires among all states in the USA Interestingly, Wyoming is observed to have received significantly reduced losses. The analysis presented in this study based on certain communities selected in the mentioned states present an antithesis to this observation. This leads to the conclusion that a generalized viewpoint of risk cannot be formulated for all communities. Each community has a unique footprint, as each is featured with unique characteristics, especially pertaining to their distinctive layout. Several valuable studies have provided insightful information on the general trends of wildfire risk, but specific analysis of individual communities is also required such that custom intervention measures can be developed, which could aid in the development general policies related to fire mitigation. The results presented in this study highlight the importance of individual community risk analysis.

Certain assumptions and limitations are made in this study. For instance, the analysis does not include the presence of fences in backyards that have been shown to have occasional impact on fire

propagation. Due to the complexity of the wildfire problem it was necessary to keep the scope of the study limited. Before the proposed framework can be used for practical purposes it might require certain modifications based on intended use. The purpose of this study is to lay the foundation for future possibilities in such an approach. We believe that a generalized quantification framework for overall risk of communities is necessary to determine critical parameters for different types of communities such that leaders, policy makers and urban planners can make informed decisions regarding intervention measures in the future.

Data accessibility. The probability of ignition data for the locations presented in this study is taken from archives of Wildland Fire Assessment System (WFAS) (https://www.wfas.net). The wind data for all locations is obtained from the archives of National Oceanic and Atmospheric Administration (NOAA) (https://www.noaa.gov). The GIS data for each location is obtained based on Openstreetmaps (https://www.openstreetmap.org). The relevant datasets and Matlab codes to extract the data can be found in the Zenodo Digital Repository: http://dx.doi.org/10.5281/zenodo.3379062 [50].

Authors' contributions. H.M. and A.C. formulated the research model, collected the data, performed the analysis and wrote the paper.

Competing interests. We declare we have no competing interests.

Funding. Funding for this study is provided, in part, through the cooperative agreement no. 70NANB15H044 between the National Institute of Standards and Technology (NIST) and Colorado State University.

Acknowledgements. The authors acknowledge Colorado State University and NIST. The contents expressed in this paper are the views of the authors and do not necessarily represent the opinions or views of NIST or the US Department of Commerce. The authors also acknowledge and appreciate all feedback and comments provided by the reviewers and by the Associate Editor Dr Quazi Hassan.

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
