## [Reviewer comments · Royal Society Open Science]

Review History

RSOS-191467.R0 (Original submission)

Review form: Reviewer 1

Is the manuscript scientifically sound in its present form?

No

Are the interpretations and conclusions justified by the results?

No

Is the language acceptable?

No

Do you have any ethical concerns with this paper?

No

Have you any concerns about statistical analyses in this paper?

No

Recommendation?

Major revision is needed (please make suggestions in comments)

Comments to the Author(s)

- Too much verbose.
- Hard to follow the readings, because section and subsections were not organized in proper sequence.
- Sections sequence supposed to be the following order for this journal: Introduction, Study area, Materials, Methods, Results, Discussion, and Conclusion.
- Inappropriate and unnecessary examples were added.
- Information here and there, and hard to correlate.
- Did not follow the structure of the journal.
- Suggesting to be precise in writing all sections, and avoid unnecessary discussion or details.
- Used an existing methodological framework, only applied in different locations, nothing noble.
-

Review form: Reviewer 2**Is the manuscript scientifically sound in its present form?**

No

Are the interpretations and conclusions justified by the results?

No

Is the language acceptable?

No

Do you have any ethical concerns with this paper?

No

Have you any concerns about statistical analyses in this paper?

No

Recommendation?

Major revision is needed (please make suggestions in comments)

Comments to the Author(s)

Thank you so much for the opportunity to review your paper, "Communicating Wildland Urban Interface Fire Risk" as submitted to Royal Society Open Science. I agree that wildfire risk in wildland-urban interface communities is a pressing problem. However, the merit of your approach and unique value of this specific paper was not clear to me upon review. How is this paper different than your previous publication of a quasi physics-based graph model? It seems to add additional sites, but no new theory? I am unfortunately not a specialist in graph theory so hopefully other reviewers will be better positioned to provide feedback on the specifications of your model.

As a non-expert it was unclear to me how well this approach at assessing risk works given that there is no information available in your model about materials or vegetation, and that you have excluded a time period (October) that typically brings the worst wildfires in CA and TX. In addition, there are numerous statements about wildfires and wildfire policy in the U.S. that are incorrect. I suggest working with a collaborator who specializes in wildfire management and policy if you would like to ground your work and paper in an accurate portrayal of wildfire management. Finally, I think it would help reviewers if you carefully proofread your paper: there are inconsistencies throughout with formatting, grammar (subject-verb agreement, passive tense), capitalization, etc. A professional editor will be helpful when revising this material.

I have some more specific comments below.

Thank you again very much.

Specific comments:

Page 1: The title (to me) implies that this paper will be about communicating to the public about wildfire risk. Should the title instead use a different word choice? Maybe assessing wildfire risk?

Page 3, line 10: this is the most catastrophic season on record in the US not just in California.

Page 3, line 29: primary aim of what? Also there has been an emphasis on fire-adapted communities, or living with fire through land use planning, home-level mitigation, emergency response, etc. See literature on fire-adapted communities or national cohesive strategy.

Page 3, line 35: "no clear standardized design policies exist" - I'm not sure what this means

Page 3, line 36: I don't think the term severe wildfire suppression is widely used in the literature; seems like it needs a definition.

Page 3, line 39: "rapid growth in wildland ecology" doesn't make sense, and I don't think that's considered the key reason for increased wildfire ignitions or damage

Page 3, line 43: check for grammar and meaning. This doesn't make sense to me: "In light of these developments, a shift in Wildfire management paradigm is required such that mitigation efforts are geared towards wildland urban interface as well. Appropriate attention needs to be placed on communicating and regulating community risk as well"

Page 4, line 1: what does "WUI risk of communities" mean? The wildfire risk in WUI communities? Also the paper inconsistently uses WUI and the full term.

Page 4, line 16: why May-September? Especially for Oakland (CA) and Austin (TX), that doesn't make sense. Fires occur in October and beyond (and TX has an earlier fire season as well in grassland areas).

Page 4, lines 27-36: this paragraph does not make sense and contains factual errors. Wildfires were considered hazards long before the 2017 legislation the authors refer to. They also say in this paragraph that fires are both like and unlike other hazards but with no citations or explanation for the apparent contradiction.

Page 4, line 40: here and elsewhere authors seem confused about the extent and nature of the WUI. As I've understood it, the WUI is primarily where homes meet and intermingle with wildlands, and WUI growth is primarily a story of housing growth, yet they focus a lot on population.

Page 4, line 43: is this the central point of the paper? Seems a bit late to be getting here: "Have we quantified the underlying factors involved in wildfire events and do we understand their importance relative to each other?"

Page 4, line 46: expenditures are also rising because of climate change, increasing WUI development, etc. The suppression policies haven't really changed much over time and yet costs have increased, so I think it's important to clarify why the costs are rising.

Page 5, line 7: grammar. Have stopped noting grammatical (verb-tense, etc) errors - please work with an editor to address issues throughout the paper.

Page 5, line 29-35: these are different definitions of exposure and vulnerability than I am familiar with. I kept expecting a mention of vulnerability = exposure + sensitivity. Where are these definitions coming from?

Page 7: why are these archetypal communities?

Page 7, line 36: what does this mean for vegetation? "Based on vegetation alone communities can be classified into different types. In order to do a comparative study between the communities, individual vegetation (trees and shrubs) are not modeled exclusively". It seems to me that they are not modeled at all. How does this relate to recent studies of why houses are lost (comparing structural choices vs. defensible space?). Why isn't that literature referenced more here? For example, Factors Associated with Structure Loss in the 2013-2018 California Wildfires, AD Syphard, JE Keeley - Fire, 2019 and similar

Page 8, line 18: Relieved to see authors are familiar with seasonal variation in wind, but why is the study period from May - September then? For example, Santa Ana winds (which the authors mention specifically) don't occur during that time period.

Page 11, line 40: the lack of information related to specific materials seems like a major limitation of this effort. How could graph theory be used to understand risk at the community level without any information on materials used in buildings, and no

Decision letter (RSOS-191467.R0)

14-Oct-2019

Dear Dr Mahmoud,

The editors assigned to your paper ("Communicating Wildland Urban Interface Fire Risk") have now received comments from reviewers. We would like you to revise your paper in accordance with the referee and Associate Editor suggestions which can be found below (not including confidential reports to the Editor). Please note this decision does not guarantee eventual acceptance.

Please submit a copy of your revised paper before 06-Nov-2019. Please note that the revision deadline will expire at 00.00am on this date. If we do not hear from you within this time then it will be assumed that the paper has been withdrawn. In exceptional circumstances, extensions may be possible if agreed with the Editorial Office in advance. We do not allow multiple rounds of revision so we urge you to make every effort to fully address all of the comments at this stage. If deemed necessary by the Editors, your manuscript will be sent back to one or more of the original reviewers for assessment. If the original reviewers are not available, we may invite new reviewers.

- Data accessibility

It is a condition of publication that all supporting data are made available either as supplementary information or preferably in a suitable permanent repository. The data accessibility section should state where the article's supporting data can be accessed. This section should also include details, where possible of where to access other relevant research materials such as statistical tools, protocols, software etc can be accessed. If the data have been deposited in an external repository this section should list the database, accession number and link to the DOI

for all data from the article that have been made publicly available. Data sets that have been deposited in an external repository and have a DOI should also be appropriately cited in the manuscript and included in the reference list.

If you wish to submit your supporting data or code to Dryad (<http://datadryad.org/>), or modify your current submission to dryad, please use the following link:
<http://datadryad.org/submit?journalID=RSOS&manu=RSOS-191467>

- **Competing interests**

- **Authors' contributions**

- **Acknowledgements**

- **Funding statement**

Kind regards,
Anita Kristiansen
Editorial Coordinator
Royal Society Open Science
openscience@royalsociety.org

on behalf of Professor Quazi Hassan (Associate Editor) and R. Kerry Rowe (Subject Editor)
openscience@royalsociety.org

Comments to Author:

Reviewers' Comments to Author:

Reviewer: 1

Comments to the Author(s)

- Too much verbose.

- Hard to follow the readings, because section and subsections were not organized in proper sequence.
- Sections sequence supposed to be the following order for this journal: Introduction, Study area, Materials, Methods, Results, Discussion, and Conclusion.
- Inappropriate and unnecessary examples were added.
- Information here and there, and hard to correlate.
- Did not follow the structure of the journal.
- Suggesting to be precise in writing all sections, and avoid unnecessary discussion or details.
- Used an existing methodological framework, only applied in different locations, nothing noble.
-

Please see PDF attached.

Reviewer: 2

Comments to the Author(s)

Thank you so much for the opportunity to review your paper, "Communicating Wildland Urban Interface Fire Risk" as submitted to Royal Society Open Science. I agree that wildfire risk in wildland-urban interface communities is a pressing problem. However, the merit of your approach and unique value of this specific paper was not clear to me upon review. How is this paper different than your previous publication of a quasi physics-based graph model? It seems to add additional sites, but no new theory? I am unfortunately not a specialist in graph theory so hopefully other reviewers will be better positioned to provide feedback on the specifications of your model.

As a non-expert it was unclear to me how well this approach at assessing risk works given that there is no information available in your model about materials or vegetation, and that you have excluded a time period (October) that typically brings the worst wildfires in CA and TX. In addition, there are numerous statements about wildfires and wildfire policy in the U.S. that are incorrect. I suggest working with a collaborator who specializes in wildfire management and policy if you would like to ground your work and paper in an accurate portrayal of wildfire management. Finally, I think it would help reviewers if you carefully proofread your paper: there are inconsistencies throughout with formatting, grammar (subject-verb agreement, passive tense), capitalization, etc. A professional editor will be helpful when revising this material.

I have some more specific comments below.

Thank you again very much.

Specific comments:

Page 1: The title (to me) implies that this paper will be about communicating to the public about wildfire risk. Should the title instead use a different word choice? Maybe assessing wildfire risk?

Page 3, line 10: this is the most catastrophic season on record in the US not just in California.

Page 3, line 29: primary aim of what? Also there has been an emphasis on fire-adapted communities, or living with fire through land use planning, home-level mitigation, emergency response, etc. See literature on fire-adapted communities or national cohesive strategy.

Page 3, line 35: "no clear standardized design policies exist" - I'm not sure what this means

Page 3, line 36: I don't think the term severe wildfire suppression is widely used in the literature; seems like it needs a definition.

Page 3, line 39: "rapid growth in wildland ecology" doesn't make sense, and I don't think that's considered the key reason for increased wildfire ignitions or damage

Page 3, line 43: check for grammar and meaning. This doesn't make sense to me: "In light of these developments, a shift in Wildfire management paradigm is required such that mitigation efforts are geared towards wildland urban interface as well. Appropriate attention needs to be placed on communicating and regulating community risk as well"

Page 4, line 1: what does “WUI risk of communities” mean? The wildfire risk in WUI communities? Also the paper inconsistently uses WUI and the full term.

Page 4, line 16: why May-September? Especially for Oakland (CA) and Austin (TX), that doesn’t make sense. Fires occur in October and beyond (and TX has an earlier fire season as well in grassland areas).

Page 4, lines 27-36: this paragraph does not make sense and contains factual errors. Wildfires were considered hazards long before the 2017 legislation the authors refer to. They also say in this paragraph that fires are both like and unlike other hazards but with no citations or explanation for the apparent contradiction.

Page 4, line 40: here and elsewhere authors seem confused about the extent and nature of the WUI. As I’ve understood it, the WUI is primarily where homes meet and intermingle with wildlands, and WUI growth is primarily a story of housing growth, yet they focus a lot on population.

Page 4, line 43: is this the central point of the paper? Seems a bit late to be getting here: “Have we quantified the underlying factors involved in wildfire events and do we understand their importance relative to each other?”

Page 4, line 46: expenditures are also rising because of climate change, increasing WUI development, etc. The suppression policies haven’t really changed much over time and yet costs have increased, so I think it’s important to clarify why the costs are rising.

Page 5, line 7: grammar. Have stopped noting grammatical (verb-tense, etc) errors – please work with an editor to address issues throughout the paper.

Page 5, line 29-35: these are different definitions of exposure and vulnerability than I am familiar with. I kept expecting a mention of vulnerability = exposure + sensitivity. Where are these definitions coming from?

Page 7: why are these archetypal communities?

Page 7, line 36: what does this mean for vegetation? “Based on vegetation alone communities can be classified into different types. In order to do a comparative study between the communities, individual vegetation (trees and shrubs) are not modeled exclusively”. It seems to me that they are not modeled at all. How does this relate to recent studies of why houses are lost (comparing structural choices vs. defensible space?). Why isn’t that literature referenced more here? For example, Factors Associated with Structure Loss in the 2013–2018 California Wildfires, AD Syphard, JE Keeley - Fire, 2019 and similar

Page 8, line 18: Relieved to see authors are familiar with seasonal variation in wind, but why is the study period from May – September then? For example, Santa Ana winds (which the authors mention specifically) don’t occur during that time period.

Page 11, line 40: the lack of information related to specific materials seems like a major limitation of this effort. How could graph theory be used to understand risk at the community level without any information on materials used in buildings, and no

Author's Response to Decision Letter for (RSOS-191467.R0)

See Appendix A.

RSOS-191467.R1 (Revision)

Review form: Reviewer 1

Is the manuscript scientifically sound in its present form?

No

Are the interpretations and conclusions justified by the results?

No

Is the language acceptable?

Yes

Do you have any ethical concerns with this paper?

No

Have you any concerns about statistical analyses in this paper?

No

Recommendation?

Reject

Comments to the Author(s)

The study is very much appreciated. I would like to thank the authors for making many changes in the manuscript that have been suggested in the first-round including adding few references and re-organizing the chapters according to the journal's structure/format. However, the manuscript is not publishable in its current form, and thus I would like to "Reject" it for the following reasons:

- There is nothing noble in the study. An existing graph theory has been used in this study, which has already been used for similar study in the past [1]. This manuscript only applied the same concept/algorithm in different geographic locations.
- Suggested to provide reference for each information, concept or argument derived from other study, and the current form of the manuscript included only few. There are too many places remain where information/arguments were placed without references. Even to demonstrate the concept of the graph theory, the manuscript used exactly the same figure (i.e., Figure 2 in this manuscript; and Figure 1 in Mahmoud and Chulahwat 2018 [1]) without even referring it.
- The methodology of the entire framework of this study is not organized or free flow to understand.
- The algorithms/equations used in this manuscript were not properly explained.
- A key parameter is the presence of vegetation pattern and its density inside the urban areas for such modelling, and thus, suggesting to use the parameter in your model. This addition would make your study publishable by applying the existing graph theory model in the current study area.

[1] Mahmoud H, Chulahwat, A. 2018. Unraveling the complexity of wildland urban interface fires. *Scientific Reports* 8, 1-12.

Good luck with the manuscript in future submission.

Decision letter (RSOS-191467.R1)

09-Dec-2019

Dear Dr Mahmoud:

Manuscript ID RSOS-191467.R1 entitled "Assessing Wildland Urban Interface Fire Risk" which you submitted to Royal Society Open Science, has been reviewed. The comments from reviewer(s) are included at the bottom of this letter.

In view of the criticisms of the reviewer(s), I must decline the manuscript for publication in Royal

Society Open Science at this time. However, a new manuscript may be submitted which takes into consideration these comments.

Please note that resubmitting your manuscript does not guarantee eventual acceptance, and that your resubmission will be subject to re-review by the reviewer(s) before a decision is rendered.

You will be unable to make your revisions on the originally submitted version of your manuscript. Instead, revise your manuscript using a word processing program and save it on your computer.

You may also click the below link to start the resubmission process (or continue the process if you have already started your resubmission) for your manuscript. If you use the below link you will not be required to login to ScholarOne Manuscripts.

*** PLEASE NOTE: This is a two-step process. After clicking on the link, you will be directed to a webpage to confirm. ***

https://mc.manuscriptcentral.com/rsos?URL_MASK=780bdf9509704987b17bb75f9b9a958e

Because we are trying to facilitate timely publication of manuscripts submitted to Royal Society Open Science, your resubmitted manuscript should be submitted by 07-Jun-2020. If you are unable to submit by this date please contact the Editorial Office for options.

I look forward to a resubmission.

Kind regards,
Anita Kristiansen
Editorial Coordinator
Royal Society Open Science
openscience@royalsociety.org

on behalf of Professor Quazi Hassan (Associate Editor) and R. Kerry Rowe (Subject Editor)
openscience@royalsociety.org

Associate Editor Comments to Author (Professor Quazi Hassan):

Comments to the Author:

Upon careful examination of the revised manuscript and reviewers perspective, I consider that the manuscript has merit but unpublishable in its current form. In case of resubmission to RSOS, I suggest to make the further following improvements:

- This manuscript is an application of the same concept/algorithm in different geographic locations. Also, in order to demonstrate the concept, the authors have used exactly the same figure (i.e., Figure 2 in the current manuscript; and Figure 1 in authors earlier publication as illustrated in Mahmoud and Chulahwat, 2018. Unraveling the complexity of wildland urban interface fires. Scientific Reports 8, 1-12.) without referring.
- Require to include references in augments ideas throughout the manuscript.
- The methodology of the entire framework of this study is unorganized. Also, the algorithms/equations used in this manuscript require proper explanations.

- One of the important factor is the presence of vegetation and its density (even inside the urban areas) for modelling wildland fire risk. Such a factor is worthy to include for potential reconsideration.

Reviewer comments to Author:

Reviewer: 1

Comments to the Author(s)

The study is very much appreciated. I would like to thank the authors for making many changes in the manuscript that have been suggested in the first-round including adding few references and re-organizing the chapters according to the journal's structure/format. However, the manuscript is not publishable in its current form, and thus I would like to "Reject" it for the following reasons:

- There is nothing noble in the study. An existing graph theory has been used in this study, which has already been used for similar study in the past [1]. This manuscript only applied the same concept/algorithm in different geographic locations.
- Suggested to provide reference for each information, concept or argument derived from other study, and the current form of the manuscript included only few. There are too many places remain where information/arguments were placed without references. Even to demonstrate the concept of the graph theory, the manuscript used exactly the same figure (i.e., Figure 2 in this manuscript; and Figure 1 in Mahmoud and Chulahwat 2018 [1]) without even referring it.
- The methodology of the entire framework of this study is not organized or free flow to understand.
- The algorithms/equations used in this manuscript were not properly explained.
- A key parameter is the presence of vegetation pattern and its density inside the urban areas for such modelling, and thus, suggesting to use the parameter in your model. This addition would make your study publishable by applying the existing graph theory model in the current study area.

[1] Mahmoud H, Chulahwat, A. 2018. Unraveling the complexity of wildland urban interface fires. Scientific Reports 8, 1-12.

Good luck with the manuscript in future submission.

Author's Response to Decision Letter for (RSOS-191467.R1)

See Appendix B.

RSOS-201183.R0

Review form: Reviewer 1

Is the manuscript scientifically sound in its present form?

Yes

Are the interpretations and conclusions justified by the results?

Yes

Is the language acceptable?

Yes

Do you have any ethical concerns with this paper?

No

Have you any concerns about statistical analyses in this paper?

No

Recommendation?

Accept with minor revision (please list in comments)

Comments to the Author(s)

- Please introduce any abbreviation/acronym when used for the first time in the manuscript. For example, WUI on page 6 Line 32 and SI on page 8 line 53. Check for other abbreviations/acronyms in the manuscript.
- Please use a uniform date format in the manuscript. For example, used 4/1/2007 format on page 13 line 10, and different in other locations. Suggesting to use: XX Month Year; or Month XX, Year format.

Decision letter (RSOS-201183.R0)

Dear Dr Mahmoud

On behalf of the Editor, I am pleased to inform you that your Manuscript RSOS-201183 entitled "Assessing Wildland Urban Interface Fire Risk" has been accepted for publication in Royal Society Open Science subject to minor revision in accordance with the referee suggestions. Please find the referees' comments at the end of this email.

The reviewers and Subject Editor have recommended publication, but also suggest some minor revisions to your manuscript. Therefore, I invite you to respond to the comments and revise your manuscript.

- Ethics statement

- Data accessibility

If you wish to submit your supporting data or code to Dryad (<http://datadryad.org/>), or modify your current submission to dryad, please use the following link:
<http://datadryad.org/submit?journalID=RSOS&manu=RSOS-201183>

- **Competing interests**

- **Authors' contributions**

- **Acknowledgements**

- **Funding statement**

Because the schedule for publication is very tight, it is a condition of publication that you submit the revised version of your manuscript before 23-Jul-2020. Please note that the revision deadline will expire at 00.00am on this date. If you do not think you will be able to meet this date please let me know immediately.

on behalf of Professor Quazi Hassan (Associate Editor) and R. Kerry Rowe (Subject Editor)
openscience@royalsociety.org

Associate Editor Comments to Author (Professor Quazi Hassan):

Associate Editor

Comments to the Author:

May I suggest the authors to address minor issue raised by the reviewer. In addition, they may include the some relevant literature that highlights the importance of vegetation in the WUI, e.g.,

Michael et al. Economic Assessment of Fire Damage to Urban Forest in the Wildland–Urban Interface Using Planet Satellites Constellation Images. *Remote Sens.* 2018, 10, 1479.

Ahmed et al. Remote sensing-based analysis of wildland fire-induced risk assessment at the community level. *Sensors* 2018, 18, 1570.

Reviewer comments to Author:

Reviewer: 1

Comments to the Author(s)

- Please introduce any abbreviation/acronym when used for the first time in the manuscript. For example, WUI on page 6 Line 32 and SI on page 8 line 53. Check for other abbreviations/acronyms in the manuscript.

- Please use a uniform date format in the manuscript. For example, used 4/1/2007 format on page 13 line 10, and different in other locations. Suggesting to use: XX Month Year; or Month XX, Year format.

Author's Response to Decision Letter for (RSOS-201183.R0)

See Appendix C.

Decision letter (RSOS-201183.R1)

Dear Dr Mahmoud,

It is a pleasure to accept your manuscript entitled "Assessing Wildland Urban Interface Fire Risk" in its current form for publication in Royal Society Open Science.

on behalf of Professor Quazi Hassan (Associate Editor) and R. Kerry Rowe (Subject Editor)
openscience@royalsociety.org

Appendix A

Reviewer 1

The authors sincerely thank the reviewer for the constructive and intellectual feedback provided and for the opportunity to reply to the comments. The paper has been carefully revised to address all issues raised. This document provides our point-by-point responses to the reviewer's comments.

Reviewer General Comments and Authors responses:

1. Too much verbose.

Response: Thank you for the comment. We agree with the reviewer. The entire manuscript has been reviewed to make sure sentences are concise and to the point.

2. Hard to follow the readings, and section and subsections are not in proper sequence.

Response: Thank you for the comment. We have revised the paper and placed sections in the proper order as noted in the journal template. We also revised the introduction so that the paragraphs are presented in more logical order.

3. Sections sequence supposed to be the following order for this journal: Introduction, Study area, Materials, Methods, Results, Discussion, and Conclusion.

Response: Thank you for the comment. We apologize for overlooking the proper section order. We have revised the order of the sections to follow the journal suggested sequence.

4. Information here and there.

Response: Thank you for the noting the issue. The entire manuscript was revisited to make sure the writing is concise and there is proper flow of information.

5. Did not follow the structure of the journal.

Response: Thank you for the comment again. We sincerely apologize and we have addressed this issue as noted above.

6. Suggesting to be precise in writing all sections, and avoid unnecessary discussion or details.

Response: Thanks a lot for the comment. We have embraced the suggestions provided by the reviewer and have revised the entire manuscript accordingly.

7. Used an existing methodological framework, only applied in different locations, nothing noble.

Response: The reviewer raised a great point that deserves further clarification. We indeed acknowledge that the framework for vulnerability has been developed and published in a recent study by us. We would like to take this opportunity to reflect on the novelty of the work, which was raised by both reviewers. In this study, the framework we previously developed has been extended and used to understand the factors that contribute to community risk by quantifying the interplay between the probability of ignition and vulnerability. We quantified and showed that the correlation between these three factors depends on the community being evaluated. Based on the results shown, we believe that this study emphasizes the need for exploring unique viable solutions to reduce risk for every community independently as opposed to embracing a generalized approach as is currently the case. Our hope is that this work, along with other studies, could form new directions towards establishing effective policies for mitigating wildfire risk to communities.

8. Degree symbol missing!

Response: Thank you for the comment. We have added the “degree” symbol.

9. The most accurate tools are arguably those that are founded on computational fluid dynamics (CFD).

Response: The authors appreciate the comment. While there is no reference that explicitly compares the different methods. The statement was made since CFD methods are based on physics as opposed to semi-physics or empirical methods. We have revised the manuscript as underlined below to clarify this issue.

“Computational fluid dynamics (CFD) models have been found to be significantly efficient for modeling wildfire propagation, since they are based on physics of the problems as opposed to semi-physics or empirical methods.”

10. This subsection could be included in the Introduction section, because such motivation triggered you to perform this study and helped to setup the objectives. Some of the information is redundant though.

Response: Thank you for the note. We agree with the reviewer’s suggestion. We have made the suggested changes and we have amended this section to the introduction.

11. “Since the focus of this study is towards community specific risk, the probability of fire reaching the interface, once initiated, is assumed to be one for all cases” *What is the use of fire behavior models then?*

Response: Thank you for the comment. There has probably been some miscommunication. The probability of fire reaching the interface is a critical part of risk assessment; however, the focus of this study is to show the importance of community-based risk analysis frameworks. While comprehensive wildland propagation models already exist and are relatively well developed, community fire propagation and risk are not. To highlight the sensitivity of different characteristics of communities to fire risk, the wildland propagation probability was kept a constant. It was chosen as 1 to represent maximum possible probability for each community. It is not that wildland propagation is not relevant it is just that for the intents and purposes of this study a constant probability was used so as to normalize the effect of wildland propagation.

12. “The definition of importance could vary depending on the focus of the analysis, but the alternative formulation could be used to account for high-value resources within a community. *Reference?*”

Response: Thank you for the comment. By varying definition of importance, we meant to say that some critical structures could have higher importance values, depending on their value to the community, which could be reflected in the weight assigned to the structure to obtain the overall vulnerability. For example, one might consider assigning higher weights to a hospital, school, highly inflammable structures and other sensitive structures. This classification, however, is subjective and could vary for different communities depending on the priorities for that particular community. Hence, a generalized representation for this concept was not presented.

13. “The mean probability is averaged over K MPPs $P(s)$ m (Eq. 3.3), which is selected as $K = 10$ for all analysis due to computational limitations.” *If you have a computational limitations, and due to that an average value was used, how could you derive an authentic results from it? Can it be reliable then?*

Response: The author raises an excellent point. And we totally agree that a computational limitation might hinder the applicability of the results. We would like to clarify that we did not go beyond a K value of higher than 10 since we found that the vulnerability values began to converge at K value of 10 i.e. the change in vulnerability values beyond that point was considerably small. This was a misinformation on our part since we meant to generalize the statement. That is to say, if the change in vulnerability values becomes a constant after certain number of iterations then convergence is assumed to be achieved. If the user wishes to use higher number of iterations (K value) then they can do so, but the computation time would increase accordingly. We have modified the sentence as underlined below to clarify this issue.

The mean probability is averaged over K MPPs $P(s)$ m (Eq. 3.3), which is selected as $K = 10$ for all analysis, since such value was sufficient to achieve convergence. It is worth noting that in general one would have to consider the trade-off between the level of convergence considered acceptable and the substantial computational demand added if higher K values are used.

14. “The details on the graph networks formulated for each community are discussed in the SI text.” *An abbreviation should be introduced before placing it further in texts. Though it's known by most of us, it should be introduced in each manuscript.*

Response: Thank you for the note. We have introduced the abbreviation prior to placing it further in the text as underlined below.

The details on the graph networks formulated for each community are discussed in the Supporting Information (SI).

15. *Why these maps are presented in this results section? The proper place for these maps are in the study area section!*

Response: Thank you for the comment. We completely agree. We have made a new section called “Study Areas”, before Materials and Methods, and we have moved the figure into that Section as suggested.

16. *Please respect the journal template. This journal does not accept Methods and Materials section at the end of the paper!*

Response: We apologize for overlooking the template. The paper was a direct transfer from Transaction A of the Royal Society, and we assumed that the format is the same. We have checked all formatting and strictly followed the journal template in the revised version.

17. “...and distance ($d^{(b,s)}$) between nodes b and s. The function is based on the ember model developed by Martin and Hillen [47]” *What is the "b" here?*

Response: Thank you for the comment. The index “b” represents the boundary nodes of the nearby wildland vegetation. We have modified the manuscript as shown below to further clarify this issue.

and distance ($d^{(b,s)}$) between nodes b and s, where node b is one of the boundary nodes of the adjacent wildland. The function is based on the ember model developed by Martin and Hillen [47].

18. “where $pk_u(d)$ and $pk_l(d)$ are appropriate upper and lower percentiles allotted on a daily basis by station managers and $vk_u(d)$ and $vk_l(d)$ are the performance index values corresponding to the percentiles selected.” *Don't get it! For ignition probability, you do need an ignition source, and you did not mention anything about it. Rather, the term you used should be the danger conditions to me.*

Response: Thank you for the comment. Yes, we agree you do need an ignition source. The wildland vegetation shown in the community layout maps are the ignition source. Each point in these vegetations has the same danger rating. Since the term - probability of fire reaching the interface, is equal to 1; therefore, each node in the corresponding community wildland vegetation have the same probability of causing ignition. In response to comment 11 we have previously mentioned why it was necessary to assume the wildland probability as 1. The terms $v_{ku}(d)$ and $v_{kl}(d)$ represent the danger rating for two different percentiles ($p_{ku}(d)$ and $p_{kl}(d)$) for a particular day. Based on these two values we could linearly map the observed danger value to a specific percentile. The danger rating is used to assess the probability but that is done by converting the observed danger rating to a specific percentile. Since the terms $p_{ku}(d)$ and $p_{kl}(d)$ are not constant and vary each day; therefore, the observed daily danger rating has to be mapped individually each time. So yes, danger rating is used in the analysis, but it is converted to a probability using the terms $p_{ku}(d)$, $p_{kl}(d)$, $v_{ku}(d)$ and $v_{kl}(d)$. To further clarify a statement has been added to the text, as shown below.

The National Digital Forecast Database for different locations across the U.S. The Fire danger forecasts result in rating levels that take into account current and antecedent weather, fuel types, and both live and dead fuel moisture. It primarily utilizes two performance indices - (1) Burning Index (BI) and (2) Energy Release Component (ERC). Assigning the fire danger index reflects staffing levels and climatological class breakpoints. Staff class represents the max/min fire danger rating of a location by assigning percentile values for the performance index selected (i.e BI or ERC) for a specific day. This fire danger rating is then utilized to calculate an ignition probability. Specifically, linear interpolation is used to determine probability of ignition ($P(Z)$), as given by Eq. 3, where $p_{ku}(d)$, $p_{kl}(d)$ are appropriate upper and lower percentiles allotted on a daily basis by station managers and $v_{ku}(d)$ and $v_{kl}(d)$ are the performance indices values corresponding to the percentiles selected.

19. "Some details on the scaling factor have been discussed in the SI." *Incomplete sentence!*

Response: The authors appreciate the comment. We have revised the sentence as underlined below.

Some details on the parameters controlling the scaling factor and the role in formulating the fire intervention framework is discussed in the SI.

Reviewer 2

General Comments:

Thank you so much for the opportunity to review your paper, “Communicating Wildland Urban Interface Fire Risk” as submitted to Royal Society Open Science. I agree that wildfire risk in wildland-urban interface communities is a pressing problem. However, the merit of your approach and unique value of this specific paper was not clear to me upon review. How is this paper different than your previous publication of a quasi physics-based graph model? It seems to add additional sites, but no new theory? I am unfortunately not a specialist in graph theory so hopefully other reviewers will be better positioned to provide feedback on the specifications of your model.

Response: The reviewer raised a great point that deserves further clarification. We indeed acknowledge that the framework for vulnerability has been developed and published in a recent study by us. We would like to take this opportunity reflect on the novelty of the work, which was raised by both reviewers. In this study, the framework we previously developed has been extended and used to understand the factors that contribute to community risk by quantifying the interplay between the probability of ignition and vulnerability. We quantified and showed that the correlation between these three factors depend on the community being evaluated. Based on the results shown, we believe that this study emphasizes the need for exploring unique viable solutions to reduce risk for every community independently as opposed to embracing a generalized approach as is currently the case. Our hope is that this work, along with other studies, could form new directions towards establishing effective policies for mitigating wildfire risk to communities.

As a non-expert it was unclear to me how well this approach at assessing risk works given that there is no information available in your model about materials or vegetation, and that you have excluded a time period (October) that typically brings the worst wildfires in CA and TX.

Response: The reviewer raises another excellent point. With respect to the absence of material and vegetation, we pointed out this issue and discussed it in the manuscript as noted below. Essentially, we normalized the features of vegetations and material in the investigated communities such that the stochastic nature of vegetation does not affect the analysis. We also would like to note that while each community has a unique footprint attributing to structure density, community layout, vegetation distribution and others. Based on vegetation alone communities can be classified into different types, as shown in figure below. Figures (a), (c) and (d) show communities with different degrees of vegetation ranging from low (Fig. (d)) to high (Fig. (a)). All communities were assumed to have the configuration shown in Fig. (b), where the community is interfaced with the wildland, but very limited vegetation exist inside the community. An example of recent fires in which the which the vegetation played a minor role in fire growth (due to

proximity of homes to each other) include the 2017 Coffey Park fire, the 2018 Euclid fire, the 2018 Fountaingrove II, and, the 2013 Mountain Shadows Fire. We have reflected further on this as noted below.

(a)

(b)

(c)

(d)

Fig. Different types of vegetation distributions among communities

“Since the focus of this study is to draw out a comparison between the selected communities, all ways in each community are assumed to be identical in nature i.e. possess same material properties.”

“Each community has a unique footprint attributing to structure density, community layout, vegetation distribution and others. Based on vegetation alone communities can be classified into different types. In order to do a comparative study between the communities, individual vegetation (trees and shrubs) are not modeled exclusively. This is done to normalize the features of all communities such that the stochastic nature of vegetation does not affect the analysis. Although the proposed model can capture presence of discrete vegetation surrounding the houses, the community layout in this study is derived from openstreetmap.org, which had no information available on the location of discrete vegetation near the houses. To circumvent this problem in the future, GIS data can be used in conjunction with image pattern recognition algorithms to identify stray vegetation, which can be overlaid to obtain accurate representation of the community data. Once the vegetation is mapped out it will be incorporated into the

community graph in the form of set of ways such that individual nodes would represent parts of vegetation and properties of vegetation would be assigned to nodes.”

With respect to including the month of October in the analysis, we agree with the reviewer that the month of October also poses a risk to communities. Also, devastating fires have taken place in the months of November and December. For example, the 2018 CampFire in CA occurred in November. With that in mind, we would like to clarify that our main goal was to show the variation in the relationship between ignition, risk, and vulnerability across different months of the year and we felt that choosing May – September is sufficient to highlight such relationships.

In addition, there are numerous statements about wildfires and wildfire policy in the U.S. that are incorrect. I suggest working with a collaborator who specializes in wildfire management and policy if you would like to ground your work and paper in an accurate portrayal of wildfire management. Finally, I think it would help reviewers if you carefully proofread your paper: there are inconsistencies throughout with formatting, grammar (subject-verb agreement, passive tense), capitalization, etc. A professional editor will be helpful when revising this material.

We sincerely appreciate this comment, and of course all comments for that matter. We would like to note, to the reviewer’s suggestion, that we are now working with policy makers and social scientists on expanding the research, so it is more specific to U.S. practice and policies. We have revised the manuscript very carefully to make sure that no false or immature statements are made.

Specific Comments:

Reviewer General Comments and Authors responses:

1. Page 1: The title (to me) implies that this paper will be about communicating to the public about wildfire risk. Should the title instead use a different word choice? Maybe assessing wildfire risk?

Response: Thank you for the comment. We see the reviewer’s point of view. We had intended to say that risk should be communicated based on community’s specific features. We have changed the title as suggested.

2. Page 3, line 29: primary aim of what? Also there has been an emphasis on fire-adapted communities, or living with fire through land use planning, home-level mitigation, emergency response, etc. See literature on fire-adapted communities or national cohesive strategy.

Response: The reviewer raises an excellent point and we agree with the question/suggestion. We have revised the paper as underlined below to eliminate the confusion.

Suppression and management of fuel build-up in wildlands has been one of the main tactics for lowering WUI risk to communities, which has alone proven to be insufficient [9–12].

3. Page 3, line 35: “no clear standardized design policies exist” – I’m not sure what this means.

Response: Thank you for the comment. By no clear standardized design policies exist, we meant to say that There are several standards and codes that were developed to address the problem of WUI fires; however, none extensively covers all aspects of the problem. There is currently no standardized method of risk assessment that can be applied nationwide to WUI communities in the U.S. [1].

[1] USDA and USDI. Protecting people and natural resources: a cohesive fuels treatment strategy, Technical report, USDA and USDI, 2006

4. Page 3, line 36: I don’t think the term severe wildfire suppression is widely used in the literature; seems like it needs a definition.

Response: The authors thank the reviewer for the comment. Wildfire suppression is a range of firefighting tactics used to suppress wildfires. We have added the definition to the manuscript as underlined below and the term severe has been substituted from the text.

Currently, wildfire suppression, which is the range of firefighting tactics used to suppress wildfires, is aimed at suppression of most fires. This has led to reduction in controlled small-scale fires, aiding in reducing wildland density and provide an ecological balance.

5. Page 3, line 39: “rapid growth in wildland ecology” doesn’t make sense, and I don’t think that’s considered the key reason for increased wildfire ignitions or damage

Response: Thank you for the comment. We would like to highlight that the statement was made to point out the increase in intensity of wildfires, and not to point out the increase in frequency of wildfires. We have already included two references that emphasize that increase in wildland density is correlated to the increase in intensity of wildfires. We agree that there are other factors, such as establishing communities near wildland is also a major factor. We have modified the manuscript to reflect this point.

In the absence of any natural reduction mechanism and given the limited fuel management strategies, however, increase in wildland density has resulted in significant increase in high intensity wildfires [10,13], not to mention certain other factors, such as establishing communities near wildlands.

6. Page 3, line 43: check for grammar and meaning. This doesn't make sense to me: "In light of these developments, a shift in Wildfire management paradigm is required such that mitigation efforts are geared towards wildland urban interface as well. Appropriate attention needs to be placed on communicating and regulating community risk as well"

Response: Thank you for the comment. We have revised the sentence as noted below.

In recognizing the major factors contributing to wildfire risk, a paradigm shift in wildfire management is required such that mitigation efforts are geared towards communities as well as the wildland.

7. Page 4, line 1: what does "WUI risk of communities" mean? The wildfire risk in WUI communities? Also the paper inconsistently uses WUI and the full term.

Response: Thank you for noting the inconsistency. We have revised the whole paper to eliminate these issues. All related terms have been replaced by the term "wildfire risk to communities".

8. Page 4, line 16: why May-September? Especially for Oakland (CA) and Austin (TX), that doesn't make sense. Fires occur in October and beyond (and TX has an earlier fire season as well in grassland areas).

Response: Thank you for the comment, as we noted to the response to the general comment above, we agree that the month of October and even beyond October also poses risk to communities. As the reviewer know, devastating fires have taken place in the months of November and December. For example, the 2018 CampFire in CA occurred in November. With that in mind, we would like to clarify that our main goal/intention was to show the variation in the relationship between ignition, risk, and vulnerability across different months of the year and we felt that choosing May – September is sufficient to highlight such a relationship.

9. Page 4, lines 27-36: this paragraph does not make sense and contains factual errors. Wildfires were considered hazards long before the 2017 legislation the authors refer to. They also say in this paragraph that fires are both like and unlike other hazards but with no citations or explanation for the apparent contradiction.

Response: Thank you for the comment, and apologies for the confusion. We did not intend on saying that is not a hazard. The 2017 bill recognized that budget allocation for suppression of catastrophic wildfires should be treated similar to other hazards. Only until this bill passed agencies like the Forest Service must borrow from non-fire accounts

when fire suppression costs exceed the budget. With the passing of this bill, this is no longer the case (<https://simpson.house.gov/issues/issue/?IssueID=120520>).

10. Page 4, line 40: here and elsewhere authors seem confused about the extent and nature of the WUI. As I've understood it, the WUI is primarily where homes meet and intermingle with wildlands, and WUI growth is primarily a story of housing growth, yet they focus a lot on population.

Response: Thank you for the clarification. We have revised the manuscript to make sure the terms are not mixed.

11. Page 4, line 43: is this the central point of the paper? Seems a bit late to be getting here: "Have we quantified the underlying factors involved in wildfire events and do we understand their importance relative to each other?"

Response: Thank you for the note. We agree with the reviewer and have moved this point earlier in the manuscript to convey the message early on.

12. Page 4, line 46: expenditures are also rising because of climate change, increasing WUI development, etc. The suppression policies haven't really changed much over time and yet costs have increased, so I think it's important to clarify why the costs are rising.

Response: Thank you for the comment. As suggested by the reviewer, the statement on page 4, line 46 has been modified as shown below.

Wildfire mitigation is primarily focused on complete fire suppression and control [22], which combined with climate change, increasing WUI development and certain other factors, ultimately results in high expenditure [23,24].

13. Page 5, line 29-35: these are different definitions of exposure and vulnerability than I am familiar with. I kept expecting a mention of vulnerability = exposure + sensitivity. Where are these definitions coming from?

Response: Thank you for the comment. We have adopted definitions that are commonly used by structural engineers in hazard assessment and mitigation. Although we recognize that these definitions vary in the literature. For example, in the context of earthquake risk, it is common to define vulnerability as reaching a specific limit state as noted in the passage below, which can be found in a report published for the Mid-America Earthquake Center (<https://www.ideals.illinois.edu/bitstream/handle/2142/9177/Report%2004-04.pdf?sequence=2&isAllowed=y>). A co-author of the report is Prof. Bruce Ellingwood, who is an authority on disaster risk and reliability.

“The vulnerability needs to be described in terms of probability of a set of given limit states being reached of a system at a given location over a given period of time (0, t). Alternatively, the vulnerability can be stated in terms of occurrence rates of the prescribed limit states.”

We used a similar definition to that use in the report noted above. More specifically the definition we used can be found in *Caribbean Handbook on Risk Management* which is produced by under the supervision of the *ACP-EU Natural Disaster Risk Reduction Program* (please see <http://www.charim.net/> and <http://www.charim.net/methodology/51>)

14. Page 7: why are these archetypal communities?

Response: Thank you for the comment. This is a great question. We wanted to select communities that represent different archetypes around the country and are located in states that are known to be prone to wildfires. To eliminate the confusion, we removed the word “archetypal” from the manuscript.

15. Page 7, line 36: what does this mean for vegetation? “Based on vegetation alone communities can be classified into different types. In order to do a comparative study between the communities, individual vegetation (trees and shrubs) are not modeled exclusively”. It seems to me that they are not modeled at all. How does this relate to recent studies of why houses are lost (comparing structural choices vs. defensible space?). Why isn’t that literature referenced more here? For example, Factors Associated with Structure Loss in the 2013–2018 California Wildfires, AD Syphard, JE Keeley - Fire, 2019 and similar.

Response: The reviewer raises a very strong point. We fully agree that some examples of recent community fires can be attributed to large vegetations in the communities. As noted in our response above, we normalized the impact of vegetation. We wanted to select communities that represent different archetypes around the country and are located in states that are known to be prone to wildfires. In our selection, we also ensured that the community layouts are different from each other so that their risk can be distinctly different. We would also like to point out that the vegetation is modeled into the graph framework. Although, the vegetation is not exclusively shown in the community layout figures, but the impact of vegetation around houses is modeled by including their effect during the graph calculation and intervention steps. The ignition transfer probabilities calculated between different houses, using Eq. 3.1, takes into account the increase in fuel volume due to the presence of vegetation around houses. And the intervention factors introduced in the intervention step take into account any reduction in the vegetation volume in the defensible space around houses.

16. Page 8, line 18: Relieved to see authors are familiar with seasonal variation in wind, but why is the study period from May – September then? For example, Santa Ana winds (which the authors mention specifically) don't occur during that time period.

Response: Thank you for the comment. As we noted to the response to the general comment above, we agree that the month of October and even beyond October also poses risk to communities. As the reviewer know, devastating fires have taken place in the months of November and December. For example, the 2018 CampFire in CA occurred in November. With that in mind, we would like to clarify that our main goal/intention was to show the variation in the relationship between ignition, risk, and vulnerability across different months of the year and we felt that choosing May – September is sufficient to highlight such a relationship. And Yes, the Santa Ana winds are primarily found in the months from October-March; however, our intention in mentioning the Santa Ana winds was to only highlight the fact that wind conditions have strong correlation to wildfire risk of communities.

17. Page 11, line 40: the lack of information related to specific materials seems like a major limitation of this effort. How could graph theory be used to understand risk at the community level without any information on materials used in buildings, and no

Response: We appreciate the insight. As we noted above, we normalize the features of vegetations and material in the investigated communities such that their variation does not affect the analysis and that the results are only dependent on wind profiles and buildings layout. Our model can indeed account for different material types. We are currently working with other at CU Boulder on specific communities to incorporate the material types into the analysis to evaluate their impact on risk. We realize that this could be a shortcoming in the analysis presented in the manuscript. We, however, would like to note that we see the analysis as an initial step towards understand community risk with more specific details.

Appendix B

Associate Editor Comments to Author (Professor Quazi Hassan) :

Upon careful examination of the revised manuscript and reviewers perspective, I consider that the manuscript has merit but unpublishable in its current form. In case of resubmission to RSOS, I suggest to make the further following improvements:

The authors sincerely thank the Prof. Hassan for the reflections and comments to improve the paper. All comments noted below have been addressed.

1. This manuscript is an application of the same concept/algorithm in different geographic locations. Also, in order to demonstrate the concept, the authors have used exactly the same figure (i.e., Figure 2 in the current manuscript; and Figure 1 in authors earlier publication as illustrated in Mahmoud and Chulawat, 2018. Unraveling the complexity of wildland urban interface fires. Scientific Reports 8, 1-12.) without referring.

Response: Thank you for the comment. We replaced Figure 2 (it is now Figure 1 due to ordering of the paper) with a new figure that is different in what is in Mahmoud and Chulawat (Scientific Reports 8, 1-12). We also substantially revised the algorithm we used and the manuscript to include the effect of vegetation distribution within the community on wildfire risk and we have added a completely new subsection titled “(e) Modeling Vegetative Fuel”. In this subsection, we explored two different approaches for modeling vegetation - (1) Explicit Vegetation Modeling (2) Simplified Vegetation Modeling. The details of both methods and their associated results are described in the main manuscript and in the Supplementary Information.

2. Require to include references in augments ideas throughout the manuscript.

Response: Thank you. We have removed all ambiguous statements and made sure that every assumption made is backed up by a reference.

3. The methodology of the entire framework of this study is unorganized. Also, the algorithms/equations used in this manuscript require proper explanations.

Response: Thank you. We have added various explanations to the algorithms and equations used and made sure every equation parameter is carefully and clearly described.

4. One of the important factor is the presence of vegetation and its density (even inside the urban areas) for modelling wildland fire risk. Such a factor is worthy to include for potential reconsideration.

Response: We appreciate the emphasis placed on the importance/presence of vegetation. We have revised the manuscript so that the spatial distribution of vegetation is included in the analysis as noted in the response to Comment 1.

Reviewer 1

The study is very much appreciated. I would like to thank the authors for making many changes in the manuscript that have been suggested in the first-round including adding few references and re-organizing the chapters according to the journal's structure/format. However, the manuscript is not publishable in its current form, and thus I would like to "Reject" it for the following reasons:

1. There is nothing noble in the study. An existing graph theory has been used in this study, which has already been used for similar study in the past [1]. This manuscript only applied the same concept/algorithm in different geographic locations.

Response: Thank you. We substantially revised the algorithm we used in the manuscript to include the effect of vegetation distribution within the community on wildfire risk and we have added a completely new subsection titled "(e) Modeling Vegetative Fuel". In this subsection, we explored two different approaches for modeling vegetation - (1) Explicit Vegetation Modeling (2) Simplified Vegetation Modeling. The details of both methods and their associated results are described in the main manuscript and in the Supplementary Information. We would like to note also that the original work by Mahmoud and Chulahwat (Scientific Reports 8, 1-12) only pertained to assessment of vulnerability. In this study, we assess community risk, which required coupling of vulnerability with ignition probabilities for different month of the year for different years. And as noted previously we changed the algorithm so that it can handle vegetation characteristics of communities.

2. Suggested to provide reference for each information, concept or argument derived from other study, and the current form of the manuscript included only few. There are too many places remain where information/arguments were placed without references. Even to demonstrate the concept of the graph theory, the manuscript used exactly the same figure (i.e., Figure 2 in this manuscript; and Figure 1 in Mahmoud and Chulahwat 2018 [1]) without even referring it.

Response: Thank you. We have removed all ambiguous statements and made sure that every assumption made is backed up by a reference.

3. The methodology of the entire framework of this study is not organized or free flow to understand.

Response: Thank you. We have reorganized the manuscript and carefully reviewed our sentences to improve clarity.

4. The algorithms/equations used in this manuscript were not properly explained.

Response: Thank you. We have added various explanations to the algorithms and equations used and made sure every equation parameter is carefully and clearly described.

5. A key parameter is the presence of vegetation pattern and its density inside the urban areas for such modelling, and thus, suggesting to use the parameter in your model. This addition would make your study publishable by applying the existing graph theory model in the current study area.

[1] Mahmoud H, Chulahwat, A. 2018. Unraveling the complexity of wildland urban interface fires. *Scientific Reports* 8, 1-12.

Response: Thank you. We substantially revised the algorithm we used in the manuscript to include the effect of vegetation distribution within the community on wildfire risk and we have added a completely new subsection titled “(e) Modeling Vegetative Fuel”. In this subsection, we explored two different approaches for modeling vegetation - (1) Explicit Vegetation Modeling (2) Simplified Vegetation Modeling. The details of both methods and their associated results are described in the main manuscript and in the Supplementary Information. We would like to note also that the original work by Mahmoud and Chulahwat (*Scientific Reports* 8, 1-12) only pertained to assessment of vulnerability. In this study, we assess community risk, which required coupling of vulnerability with ignition probabilities for different month of the year for different years. And as noted previously we changed the algorithm so that it can handle vegetation characteristics of communities.

Appendix C

Associate Editor Comments

On behalf of the Editor, I am pleased to inform you that your Manuscript RSOS-201183 entitled "Assessing Wildland Urban Interface Fire Risk" has been accepted for publication in Royal Society Open Science subject to minor revision in accordance with the referee suggestions. Please find the referees' comments at the end of this email.

The reviewers and Subject Editor have recommended publication, but also suggest some minor revisions to your manuscript. Therefore, I invite you to respond to the comments and revise your manuscript.

May I suggest the authors to address minor issue raised by the reviewer. In addition, they may include some relevant literature that highlights the importance of vegetation in the WUI, e.g.,

- Michael et al. Economic Assessment of Fire Damage to Urban Forest in the Wildland–Urban Interface Using Planet Satellites Constellation Images. *Remote Sens.* 2018, 10, 1479.
- Ahmed et al. Remote sensing-based analysis of wildland fire-induced risk assessment at the community level. *Sensors* 2018, 18, 1570.

Response: The authors thank the editor for the additional comments. The above-mentioned references have been added to section ‘Modeling Vegetative Fuel’ of the main text on Page 8.

Reviewer 1 Comments

1. Please introduce any abbreviation/acronym when used for the first time in the manuscript. For example, WUI on page 6 Line 32 and SI on page 8 line 53. Check for other abbreviations/acronyms in the manuscript.

Response: Thank you for the comment. The full form of WUI has been introduced on Paragraph 2 of Page 2. Similarly, the full form of SI has been introduced on Paragraph 4 of Page 4. The main text has been checked for other acronyms as well.

2. Please use a uniform date format in the manuscript. For example, used 4/1/2007 format on page 13 line 10, and different in other locations. Suggesting to use: XX Month Year; or Month XX, Year format.

Response: As per reviewer’s suggestion, the above-mentioned statement has been modified on Paragraph 2 on Page 9 as shown below to maintain consistency throughout the main text.

“The wind speed and direction for the tests are chosen based on wind conditions of 1st May 2007 for each location.”